# Learned Cardinality Estimation under Query and Data Distribution Drift

## Abstract

The problem of estimating the cardinality of queries is central to database systems. Recently, there has been growing interest in applying machine learning to this task. However, real-world databases are dynamic: the underlying data evolves and query patterns change over time. A key limitation of existing learning-based approaches is their susceptibility to drift. To the best of our knowledge, no prior method provides provable performance guarantees in fully dynamic environments. In this paper, we design an online learner that can, by passively observing queries and their corresponding cardinalities, maintain an effective model with strong performance guarantees even under continuous distributional drift. The algorithm applies to a broad class of queries, including orthogonal range-queries and distance-based queries commonly used in practice. Our work demonstrates that effective cardinality estimation in a dynamic setting possible even without direct access to the dataset.

Beyond our algorithmic results, we establish foundational results on the learnability of distribution-based models in static and dynamic environments. Such models are valued for their interpretability and inherent robustness to drift, making them especially important in practice.

## 1 Introduction

Estimating the cardinality of a database query, i.e., the number of tuples in a dataset that satisfy the query predicate, is a fundamental problem in databases (Lipton et al., 1990). Query optimizers depend on accurate estimates of query cardinalities to choose good execution plans, and over the last decade (Ding et al., 2024), there has been increasing interest on using machine learning (ML) for this task (Wang et al., 2021). In this paper, we focus on the *query-driven* regime, where the learner learns a regression model for cardinality estimation from past queries and their cardinalities (Kipf et al., 2019; Dutt et al., 2019; Park et al., 2020; Wu et al., 2025; Hu et al., 2022).

In the real world, most databases are dynamic. Both the query distribution (which regions of the data space are queried) and the data distribution (the state of the table itself) drift over time. When queries move into unseen regions or when the data distribution shifts significantly, the performance of learned cardinality estimators is known to degrade (Negi et al., 2023; Wu et al., 2025). While there exist ML-based methods with performance guarantees under limited drift (Wu et al., 2025; Hu et al., 2022), to the best of our knowledge, none offer guarantees in a fully dynamic environment. In an orthogonal line of work, Zeighami & Shahabi (2024a;b) characterize when learned methods can succeed, including under drifting conditions, but do not prescribe concrete learning algorithms.

This paper considers the setting where both query and data distributions may evolve over time. The learner only has access to query-cardinality pairs obtained by *passively* observing the database. For each new query, it produces an estimate using its current model; once the query is executed, the true cardinality is revealed. The learner may use this information to improve its model but must also be efficient in doing so. We note that systems for handling drift—e.g., Li et al. (2022); Negi (2024); Wu & Ives (2024)—typically have much more information at their disposal, including direct access to the dataset and update sequence, as well as the ability to *actively* generate additional queries. In contrast, we pose the following question: *Is it possible to design an online learning algorithm that can maintain an effective and efficient cardinality estimator solely by passively observing user-generated queries?*

**Our contributions.** We establish that the answer to the above question is indeed affirmative under some natural conditions. Specifically, we make the following contributions:

1. We formalize cardinality estimation under drift as an online-learning problem, where the learner observes a sequence of query–cardinality pairs, each drawn from an evolving distribution induced by drifting queries and data (Section 2).

2. We propose an online learning algorithm (Section 3), *Dynamically Updated Support Set* (DUSS), which maintains a distribution-based model of the underlying dataset solely from query–cardinality feedback. DUSS supports a broad class of queries, including all standard geometric range queries (boxes, balls, halfspaces, etc.).

3. We prove that when both data and query distributions drift gradually, DUSS guarantees small expected error on each new prediction (Section 4). Furthermore, even if the query distribution changes arbitrarily, as long as the data distribution remains stable, DUSS ensures that the number of times its error exceeds a threshold is bounded. Together, these conditions cover many practical situations.

4. Beyond DUSS, we establish foundational results on the learnability of distribution-based models in static and dynamic environments (see Theorems 2.1 and 4.3). Such models are widely used in database systems; their interpretability and inherent robustness to drift make them especially valuable in practice.

5. We implement a prototype of DUSS and compare it with other methods (Section D). Across diverse settings, DUSS fulfills its provable guarantees while consistently outperforming baselines in both accuracy and efficiency.

**Related Work.** There is extensive work on cardinality estimation in the database community; a comprehensive review is beyond the scope of this paper. Here we briefly discuss the work most closely related to ours; see Appendix E for a more detailed discussion. Despite extensive work on learned cardinality estimation, techniques with provable performance guarantees are limited. Hu et al. (2022) showed that distribution-based models are PAC-learnable with sample complexity bounds. Wu et al. (2025) extended this to hypothesis classes defined via signed measures (Stein & Shakarchi, 2005), obtaining the same order of sample complexity and showing robustness under limited query drift. Both works proposed concrete learners from query–cardinality pairs: Hu et al. (2022) designed a learner that reconstructs a distributional representation of the dataset by solving a quadratic program, while Wu et al. (2025) showed a neural-network–based learner and proved that the network maintains a signed measure, thereby enjoying theoretical guarantees. However, their results do not extend to the fully dynamic setting, where both query and data distributions evolve.

The only other theoretical study of learned cardinality estimation under drift is by Zeighami & Shahabi (2024b). They showed existential results under the framework of *distribution learnability*. These results are not comparable to ours for several reasons. For example, they assume access to data updates while our framework is purely based on observing workload queries with no direct access to data. Moreover, Zeighami & Shahabi (2024b) did not explicitly design a learner, while we provide a concrete algorithm with provable guarantees for a broad class of queries, including all standard geometric queries. Complementary to these results, Zeighami & Shahabi (2024a) established lower bounds on the model size necessary for cardinality estimation.

## 2 THE LEARNING MODEL

A *range space* $\Sigma = (\mathsf{X}, \mathsf{R})$ consists of a *universe* of objects $\mathsf{X}$ and a family of subsets $\mathsf{R} \subseteq 2^{\mathsf{X}}$ called *ranges*. For example, if $\mathsf{X} = \mathbb{R}^d$, then $\mathsf{R}$ may be the set of all boxes, balls, or halfspaces in $\mathbb{R}^d$. In our context, $\mathsf{X}$ is the domain of a dataset, and $\mathsf{R}$ corresponds to families of queries, such as orthogonal range queries (boxes), distance-based queries (balls), or linear-inequality queries (halfspaces). We model a *dataset* $\mathbf{C}$ as a finite multiset of tuples from $\mathsf{X}$. For a query range $R \in \mathsf{R}$, its *cardinality* on $\mathbf{C}$ is $|\mathbf{C} \cap R|$, the number of tuples in the dataset contained in $R$.

A *data distribution* over $\Sigma$ is a probability distribution over $\mathsf{X}$. For a data distribution $D$, the corresponding *selectivity function* with respect to $\Sigma$, denoted $\mu_D : \mathsf{R} \to [0, 1]$, is defined by $\mu_D(R) = \Pr_{x \sim D}[x \in R]$, i.e., the probability that a random point drawn from $D$ lies in $R \in \mathsf{R}$. For a dataset $\mathbf{C}$, if we define the distribution $D_{\mathbf{C}}$ to be $1/|\mathbf{C}|$ for all points in $\mathbf{C}$ and 0 otherwise, then

for any query $R$, $|\mathbf{C} \cap R| = |\mathbf{C}| \cdot \mu_{D_{\mathbf{C}}}(R)$, so the cardinality-estimation problem is a special case of the selectivity-estimation problem.

In this paper, we study the general problem of learning selectivity functions over range spaces in a dynamic environment where the query and data distributions drift over time. The learner has no access to the underlying data distribution and must rely only on *observations* of the form $z = (R, s)$, i.e., query–selectivity pairs in $\mathsf{Z} = \mathsf{R} \times [0, 1]$ to learn the selectivity function. For clarity, we first present the problem in the static setting and then extend it to the dynamic setting.

## 2.1 LEARNING IN A STATIC SETTING

We model the static setting by assuming a fixed *query distribution* $Q$ over $\mathsf{R}$ and a fixed *data distribution* $D$ over $\mathsf{X}$. This corresponds to the fact that the query patterns are stable and the dataset is fixed. The learner receives observations of the form $z = (R, s)$, where $z$ is a query–selectivity pair in $\mathsf{Z}$. An observation is generated by first sampling a query $R \sim Q$ and then setting $s = \mu_D(R)$. We call the distribution of $z$, induced jointly by $Q$ and $D$, the *state distribution* (SD) $W$ over $\mathsf{Z}$.

For a *hypothesis* $h : \mathsf{R} \rightarrow [0, 1]$, define the loss[1] on an observation $z = (R, s)$ as $\ell_h(z) := |h(R) - s|$. The *expected error* of $h$ with respect to $W$ is $\mathrm{err}_W(h) = \int_z \ell_h(z) \, W(z) \, \mathrm{d}z$. Let $\mathcal{H}$ be a collection of hypotheses and let $\varepsilon \in (0, 1)$ be a tolerance parameter for acceptable error. Informally, the goal is to design a learner such that, for any $W$, it can (with high probability) learn from a finite number of observations drawn from $W$ a hypothesis $h \in \mathcal{H}$ satisfying $\mathrm{err}_W(h) \leq \inf_{h' \in \mathcal{H}} \mathrm{err}_W(h') + \varepsilon$. If such a learner exists for $\mathcal{H}$, the class is said to be $\varepsilon$-*learnable* (see Haussler (1992) for the formal definition). The number of observations required is called the *sample complexity*.

**Our results.** Our main contribution is to establish improved sample complexity bounds for *distribution-based hypothesis sets*. Suppose $\mathsf{X} \subseteq \mathbb{R}^d$ and $\mathsf{R}$ corresponds to geometric objects of constant size such as boxes, balls, or halfspaces (an arbitrary convex polygon is not of constant size). Let $\mathcal{D}$ be a family of probability distributions over $\mathsf{X}$ (e.g., histograms, mixture models, or probabilistic graphical models). Define the hypothesis set $\mathcal{M} := \mathcal{M}_{\Sigma, \mathcal{D}} = \{\mu_D \mid D \in \mathcal{D}\}$; that is, $\mathcal{M}$ is the class of selectivity functions induced by distributions in $\mathcal{D}$. For example, if $\mathcal{D}$ is the family of all histograms on $\mathsf{X}$, then $\mathcal{M}$ corresponds to the class of selectivity functions defined by histograms. We obtain the following.

**Theorem 2.1.** *Let $\Sigma = (\mathsf{X}, \mathsf{R})$ be a range space, where $\mathsf{X} \subseteq \mathbb{R}^d$ and let $\mathsf{R}$ correspond to geometric objects of constant size such as boxes, or balls or halfspaces. Let $\mathcal{D}$ be a family of probability distribution over $\mathsf{X}$ and let $\mathcal{M} := \mathcal{M}_{\Sigma, \mathcal{D}}$ be the correspondning family of selectivity functions. Then, $\mathcal{M}$ is $\varepsilon$-learnable with sample complexity $O(d^2 \, \varepsilon^{-2} (\log^4 \varepsilon^{-1}))$ for any $\varepsilon \in (0, 1)$.*

Our result improves upon the previously best-known bound of $O(\varepsilon^{-d-2} \, \mathrm{polylog}(\varepsilon^{-1}))$ (Hu et al., 2022; Wu et al., 2025). In fact, our result holds for a more general setting as stated in Theorem C.1.

## 2.2 LEARNING IN A DYNAMIC SETTING

In a dynamic environment, both the query distribution and the data distribution may evolve over time, as query patterns shift and the underlying data itself changes. To capture drift, we allow the SD to vary with time. Formally, we assume that observations are drawn from a sequence of SD's $\mathcal{W} = \langle W_1, W_2, \ldots \rangle$, where the $t$-th observation $z_t = (R_t, s_t)$ is sampled from $W_t$. Furthermore, we assume that $\mathcal{W}$ is *realizable*; i.e., there exist underlying query and data distributions $Q_t$ and $D_t$ such that $R_t \sim Q_t$ and $s_t = \mu_{D_t}(R_t)$. Intuitively, $Q_t$ describes how the $t$-th query is generated, while $D_t$ represents the data distribution against which the query is evaluated. The sequences $\langle Q_1, Q_2 \ldots \rangle$ and $\langle D_1, D_2 \ldots \rangle$ capture the evolution of the query and data distributions, respectively.

Let $\mathcal{H} \subseteq \{\mathsf{R} \rightarrow [0, 1]\}$ be a hypothesis set. In a dynamic environment, any fixed hypothesis will quickly become obsolete, so learning a single hypothesis no longer suffices. Instead, we adopt an *online learning* framework, where the learner must produce a sequence of hypothesis $\langle h_1, h_2, \ldots \rangle$: for each $t$, upon seeing the prefix $\langle z_1, \ldots, z_t \rangle$ of the observations, the learner produces a function $h_t \in \mathcal{H}$ to be used for predicting the selectivity for the next range $R_{t+1}$; the predicted selectivity can then be compared with the observation $z_{t+1}$. In other words, the learner repeatedly predicts

---

[1]Other loss functions, such as squared loss, can also be used; we adopt absolute loss here for simplicity.

the selectivity for each incoming query, receives feedback in the form of the true selectivity after query execution, and then subsequently updates its model. We consider two natural objectives for an online learner over a sequence of SDs $\mathcal{W} = \langle W_1, W_2, \ldots \rangle$. Let $\varepsilon \in [0, 1]$ be the error threshold.

1. *Tracking:* Ideally, we want the learner to ensure that the *current hypothesis* $h_t$ always gives good prediction for the next query. That is, the expected error of using $h_t$ with respect to the next SD is small: i.e., $\text{err}_{W_{t+1}}(h_t) \leq \varepsilon$ for every $t > 0$.

2. *Low regret:* Instead of insisting on the accuracy of every prediction, we want the learner to not incur too many big errors over time. Formally, we define the overall regret (up to time $t$) as the sum $\sum_{i=1}^{t} \mathbb{1}\left[\ell_{h_i}(z_i) > \varepsilon\right]$, where $\mathbb{1}[\cdot]$ returns 1 if the input condition holds or 0 otherwise. Ideally, we want to keep the overall regret low for any $t > 0$.

In general, if the drift between consecutive SDs can be arbitrarily large, it would be impossible to obtain any guarantees. Hence, we propose reasonable conditions under which the above objectives can be achieved.

**Discrepancy.** To measure how much the environment has changed, we adopt a hypothesis-class dependent notion called *discrepancy*; see Mohri & Medina (2012). For two SD's $W, W'$ over $\mathsf{Z} = \mathsf{R} \times [0, 1]$, the discrepancy is defined as $\text{disc}_{\mathcal{H}}(W, W') = \sup_{h \in \mathcal{H}} |\text{err}_W(h) - \text{err}_{W'}(h)|$. Intuitively, $\text{disc}_{\mathcal{H}}(W, W')$ measures how much a change in the underlying SD from $W$ to $W'$ affects the learner's view. For example, suppose $W$ and $W'$ are induced by $(Q, D)$ and $(Q', D')$. If $D \neq D'$ but $Q = Q'$ and the queries only touch regions unchanged between the two data distributions, then $\text{disc}(W, W') = 0$, since the change has no effect from the learner's perspective.

**Our results.** We propose a novel algorithm called DUSS (Section 3) for the dynamic setting, with good provable guarantees on its performance. Suppose $\mathsf{X} \subseteq \mathbb{R}^d$ and $\mathsf{R}$ corresponds to standard geometric objects such as boxes, balls, or halfspaces. Consider the family $\mathcal{D}$ of all *discrete distributions* over $\mathsf{X}$ and the family $\mathcal{M} := \mathcal{M}_{\Sigma, \mathcal{D}}$ of all selectivity functions with respect to $\mathcal{D}$. Let $\mathcal{W} = \langle W_1, W_2, \ldots \rangle$ be a sequence of SDs, where each $W_t$ is induced by a query distribution $Q_t$ and a data distribution $D_t$. DUSS accepts a parameter $\varepsilon \in [0, 1]$ and processes a sequence of observation $\langle z_1, z_2, \ldots \rangle$, where each $z_i \sim W_i$. At the beginning of step $t$, it has a selectivity function $\mu_{t-1} \in \mathcal{M}$. After processing each $z_t$, it updates its hypothesis from $\mu_{t-1}$ to $\mu_t$, based on $\ell_{\mu_{t-1}}(z_t)$. It maintains the following guarantees.

1. **Tracking under gradual drift (Theorem 4.5).** If both the query and data distribution can change, but the drift between any two consecutive SDs is small, i.e., $\text{disc}(W_t, W_{t+1}) = o(\varepsilon^3)$ for every $t$, then DUSS ensures that for every $t$, it produces $\mu_t \in \mathcal{M}$ such that $\text{err}_{W_{t+1}}(\mu_t) \leq \varepsilon$.

2. **Low regret under stable data but arbitrary query drift (Theorem 4.2).** If the data distribution remains "stable" (a notion we will formalize later), even if the query distribution changes arbitrarily, DUSS guarantees low regret: i.e., for any $t > 0$, $\sum_{i=1}^{t} \mathbf{1}_{\varepsilon}\left(\ell_{\mu_i}(z_{i+1})\right)$ is $O(\varepsilon^{-3})$. Moreover, DUSS only needs to update its hypothesis $O(\varepsilon^{-3})$ times.

In other words, when both query and data distributions evolve gradually, DUSS gives good predictions consistently. Even if the query distribution drifts arbitrarily, DUSS still keeps regret low as long as the data distribution remains stable. Arbitrary drift in the data distribution is hostile to any passive learner without access to the data, but in large databases with row-level updates such events between consecutive queries are rare. Beyond DUSS, we also prove that the same tracking guarantees hold for any online learner that maintains a hypothesis in $\mathcal{M}_{\Sigma, \mathcal{D}}$ with $\varepsilon$-error on the most recent $O(d^2 \varepsilon^{-2} \text{polylog}(1/\varepsilon))$ observations, extending our static guarantees naturally to the dynamic setting (Theorem 4.3).

## 3 DUSS: ONLINE SELECTIVITY LEARNING ALGORITHM

Let $\Sigma = (\mathsf{X}, \mathsf{R})$ be a range space. Our algorithm DUSS (*Dynamically Updated Support Set*) handles general range spaces; for simplicity assume $\mathsf{X} \subseteq \mathbb{R}^d$ and $\mathsf{R}$ are geometric objects (boxes, balls, halfspaces). Recall $\mathsf{Z} = \mathsf{R} \times [0, 1]$. DUSS maintains a discrete distribution $\widehat{D}$ over $\mathsf{X}$ as its model, updated on a stream of observations $\mathcal{Z} = \langle z_1, z_2, \ldots \rangle$ from $\mathsf{Z}$. Let $\mathcal{Z}_t = \langle z_1, \ldots, z_t \rangle$, and $\mathcal{Z}_{t,k}$ denote the suffix of $\mathcal{Z}_t$ of length $\min\{t, k\}$.

**Overview.** DUSS accepts an error threshold $\varepsilon$ and a window size $m \geq 0$. It maintains a weighted support $\widehat{D}$ over $\mathsf{X}$ and a selectivity function $\mu_{\widehat{D}}$. At time $t$, given $z_t = (R_t, s_t)$, the algorithm treats $\mathcal{Z}_{t,m}$ as a training set and aims to maintain the invariant $|\mu_{\widehat{D}}(R_i) - s_i| \leq \varepsilon$ for all $(R_i, s_i) \in \mathcal{Z}_{t,m}$. If $\mu_{\widehat{D}}(R_t)$ under- or overestimates $s_t$, DUSS adjusts weights inside or outside $R_t$ until balanced. If $\widehat{D}$ drifts too much, it revisits $\mathcal{Z}_{t,m}$ to restore the invariant, or resets entirely when a large data shift is detected. The pseudocode appears in Algorithm 1; below we describe the information and parameters it maintains, the weight-update rule, and the revisit/reset steps.

DUSS stores $\mathcal{Z}_{t,m}$ and maintains a candidate support $\mathsf{S} \subseteq \mathsf{X}$. If $\mathsf{X}$ is finite, $\mathsf{S} = \mathsf{X}$; if $\mathsf{X} = [0,1]^d$, $\mathsf{S}$ may be a large random sample or grid. We assume $\mathsf{S}$ has enough representational power (formalized later). Each $p \in \mathsf{S}$ has an integer weight $\omega(p)$, initially 1. Let $W_{\text{curr}} = \sum_{p \in \mathsf{S}} \omega(p)$, so $\widehat{D} = \{(p, \omega(p)/W_{\text{curr}}) \mid p \in \mathsf{S}\}$.

**Weight-update.** Given $z_t = (R_t, s_t)$, let $R_t$ be *balanced* if $|\mu_{\widehat{D}}(R_t) - s_t| \leq \varepsilon$, *light* if the estimate is too small, and *heavy* if too large. If balanced, nothing is done. Otherwise:

- If light: set $\chi = \frac{\varepsilon^2/4}{s_t - \varepsilon/2}$ and repeatedly update $\omega(p) \leftarrow (1 + \chi)\omega(p)$ for all $p \in \mathsf{S} \cap R_t$ until balanced.

- If heavy: set $\chi = \frac{\varepsilon^2/4}{1 - s_t - \varepsilon/2}$ and repeatedly update $\omega(p) \leftarrow (1 + \chi)\omega(p)$ for all $p \in \mathsf{S} \setminus R_t$ until balanced.

We track COUNT, the number of updates, which is used to trigger resets.

**Revisiting the window and Resetting.** Weight-updates may break accuracy for past queries. We check whether $W_{\text{curr}}$ has grown by more than a factor $1/(1 - \varepsilon/2)$ since initialization or the last revisit. If so, we sequentially process $\mathcal{Z}_{t,m}$, applying weight-updates to any light or heavy observation. If $W_{\text{curr}}$ again grows too much, we repeat. We prove in Section 4.1 that this always converges and the number of updates remains bounded when the data distribution is stable.

If the data distribution drifts significantly, incremental updates fail. From our analysis, if the data is stable then COUNT $\leq \tau_{\text{res}} = 16\varepsilon^{-3} \ln |\mathsf{S}|$. Thus, when COUNT $> \tau_{\text{res}}$, DUSS resets: discarding all weights and restarting from $\mathcal{Z}_{t,m}$.

## 4 ANALYSIS OF DUSS

Let $\Sigma = (\mathsf{X}, \mathsf{R})$ be a range space. Before proceeding with the analysis of DUSS, we introduce the concept of *VC-dimension*, a standard measure of the *combinatorial complexity* of a range space. The VC-dimension of $\Sigma$, denoted VC-dim$(\Sigma)$, is the size of the largest $Y \subseteq \mathsf{X}$ such that $\{R \cap Y : R \in \mathsf{R}\} = 2^Y$; if no such bound exists then VC-dim$(\Sigma) = \infty$. For example, when $\mathsf{X} = \mathbb{R}^d$ and $\mathsf{R}$ is the set of boxes, balls, or halfspaces, the VC-dimension is $2d$, $d + 2$, or $d + 1$, respectively. By contrast, if $\mathsf{R}$ is the set of convex polygons, VC-dim$(\Sigma) = \infty$. The guarantees in this section hold when VC-dim$(\Sigma)$ is bounded.

Let $\mathcal{D}$ the class of discrete distributions over $\mathsf{X}$, and $\mathcal{M} := \mathcal{M}_{\Sigma, \mathcal{D}}$ the corresponding class of selectivity functions. Recall that DUSS processes a sequence of observations $\mathcal{Z} = \langle z_1, z_2, \ldots \rangle$, where each $z_t \in \mathsf{Z} = \mathsf{R} \times [0,1]$. We analyze DUSS under the assumption that $\mathcal{Z}$ is generated from a sequence of SDs $\mathcal{W} = \langle W_1, W_2, \ldots \rangle$: i.e. for each $t$, $z_t \sim W_t$, and each $W_t$ is *realized* by a query distribution $Q_t$ over $\mathsf{R}$ and a data distribution $D_t$ over $\mathsf{X}$, so that $z_t = (R_t, s_t)$ is obtained by first sampling $R_t \sim Q_t$ and then setting $s_t = \mu_{D_t}(R_t)$. We emphasize that both $D_t$ and $Q_t$ may change over time: i.e. $D_t \neq D_{t+1}$ and $Q_t \neq Q_{t+1}$ in general. Since the hypotheses learned by DUSS during its execution are probability distributions over a fixed support set $\mathsf{S} \subseteq \mathsf{X}$, it is intuitively clear that, for DUSS to be effective, $\mathsf{S}$ must possess sufficient representational power to accurately model the evolving data distribution in $\mathcal{W}$. We formalize this requirement as follows.

**$\rho$-representative support.** For a range space $\Sigma = (\mathsf{X}, \mathsf{R})$ and a distribution $D$ over $\mathsf{X}$, a finite set $\mathcal{A} \subseteq \mathsf{X}$ is called an $\epsilon$-*sample* (or $\epsilon$-*approximation*) with respect to $D$ if, for every range $R \in \mathsf{R}$, $\left| \mu_D(R) - \frac{|R \cap \mathcal{A}|}{|\mathcal{A}|} \right| \leq \epsilon$. It is known that if VC-dim$(\Sigma) = d$, then an $\epsilon$-sample of size $O\left(\frac{d}{\epsilon^2} \log \frac{1}{\epsilon}\right)$ always exists and can be obtained easily via random sampling (Vapnik & Chervonenkis, 1971). We call a finite subset $\mathsf{S} \subseteq \mathsf{X}$ an $\rho$-*representative* with respect to a realizable sequence

$\mathcal{W} = \langle W_1, W_2, \ldots \rangle$ of SD's if, for every $t$, there exists a sub-multiset $\mathsf{S}_t \subseteq \mathsf{S}$ that is a $\rho$-sample of $\Sigma$ w.r.t. $D_t$. The choice and implementation of $\rho$-representative supports depends on the range space $\Sigma$. In Section 5, we describe a heuristic for maintaining $\mathsf{S}$ for geometric ranges. In the following analysis, we assume that $\mathsf{S}$ is a $\rho$-representative support of $\mathcal{W}$ with $\rho = c\varepsilon$ for some sufficiently small constant $c > 0$. Here $\varepsilon$ denotes the error tolerance parameter of the algorithm.

### 4.1 STABLE DATA AND ARBITRARILY DRIFTING QUERIES

We first analyze the case where the query distribution may drift arbitrarily while the data distribution remains fixed (later we relax this to a "stable" data distribution). Formally, for every $W_i \in \mathcal{W}$, we assume $D_i = D^*$, a fixed distribution over $\mathsf{X}$. Thus each observation $z_t = (R_t, s_t)$ satisfies $s_t = \mu_{D^*}(R_t)$. In contrast, the query distribution may vary freely, i.e., $Q_t$ can differ arbitrarily from $Q_{t+1}$, so $W_{t+1}$ may differ from $W_t$. Our focus here is the *low-regret* objective introduced in Section 2.

Let $\mathcal{H} \subseteq \{h : \mathsf{R} \to [0, 1]\}$ be a hypothesis class, $\mathcal{Z} = \langle z_1, z_2, \ldots \rangle$ an observation sequence, and $\varepsilon > 0$ a tolerance. Let ALG be an online learner producing $h_t \in \mathcal{H}$ after processing $z_t$. For $t \geq 0$, define $f_{\mathcal{Z}}(t, \varepsilon) = \sum_{i=1}^{t} \mathbf{1}[\ell_{h_i}(z_{i+1}) > \varepsilon]$, i.e., the number of observations in $\mathcal{Z}_t$ with error above $\varepsilon$. We say ALG has regret bound $f(t, \varepsilon)$ w.r.t. $\mathcal{W}$ if $\max_{\mathcal{Z} \sim W} f_{\mathcal{Z}}(t, \varepsilon) \leq f(t, \varepsilon)$ for all $t \geq 0$. In the following lemma, we bound the number of times the weight-update step in DUSS is triggered; see Appendix A for a proof.

**Lemma 4.1.** *Let $\mathcal{W}$ be a realizable SD sequence where the data distribution is fixed. For any observation sequence $\mathcal{Z} \sim \mathcal{W}$, DUSS performs the weight-update step at most $\tau_{res}(\varepsilon) := O(\varepsilon^{-3} \cdot \log |\mathsf{S}|)$ times, irrespective of the window size $m$.*

Since a weight-update step is only triggered if a new observation is *light* or *heavy*, i.e. DUSS's prediction on the observation is off by at least $\varepsilon$, this immediately implies that cumulative regret is bounded by $O(\varepsilon^{-3} \log |\mathsf{S}|)$. A straightforward calculation shows that DUSS performs at least $\Omega(\varepsilon^{-1})$ weight-update steps between two consecutive revisit steps, and therefore it revisits the sliding window at most $O(\varepsilon^{-2} \ln |\mathsf{S}|)$ times. As in Lemma 4.1, this bound holds independently of the window size $m$.

Next, recall that DUSS maintains a probability distribution $\widehat{D}$. Let $\widehat{D}_t$ denote that state of $\widehat{D}$ after processing $z_t$. We make the following observations: 1) after DUSS finishes processing $z_t$, $R_t$ is neither light nor heavy by design; and 2) between any two revisit steps, the selectivity of every range with respect to $\mu_{\widehat{D}}$ can change by at most $\varepsilon/2$. Combining these facts with Lemma 4.1 implies that, for any window size $m > 0$ and any observation sequence $\mathcal{Z} \sim \mathcal{W}$, DUSS ensures that

$$\max_{z \in \mathcal{Z}_{t,m}} \operatorname{err}(\mu_{\widehat{D}_t}, z) \leq 2\varepsilon. \tag{1}$$

This implies the following property (which will also be useful in Section 4.2):

**Sliding-window ERM property.** Let $\mathcal{H} \subseteq \{\mathsf{R} \to [0, 1]\}$ be a hypothesis set. Given an error threshold $\varepsilon \geq 0$ and a window parameter $m \in \mathbb{N}$, we say that an online learning algorithm ALG satisfies the $(\varepsilon, m)$-*sliding window empirical risk minimizer* property, or $(\varepsilon, m)$-window ERM property for short, with respect to an observation sequence $\mathcal{Z} = \langle z_1, z_2, \ldots, \rangle$, if for any $t$, after processing $z_t$, ALG maintains a hypothesis $h_t \in \mathcal{H}$ such that

$$\sum_{z \in \mathcal{Z}_{t,m}} \operatorname{err}(h_t, z) \leq \inf_{h \in \mathcal{H}} \sum_{z \in \mathcal{Z}_{t,m}} \operatorname{err}(h, z) + \varepsilon\, m.$$

Note that this guarantee is retrospective, as it evaluates the performance of the current hypothesis $h_t$ on the last $m$ observations. In simple terms, it implies that the total error incurred on the most recent $m$ observations is within $\varepsilon m$ of the minimum possible. As discussed earlier, assuming a fixed data distribution, DUSS maintains Inequality (1), which is a stronger condition that implies the $(\varepsilon, m)$-window ERM property. Putting everything together, we obtain the following theorem.

**Theorem 4.2.** *Let $\mathcal{W}$ be any realizable SD sequence where the data distribution is fixed. DUSS achieves a regret bound of $O(\varepsilon^{-3} \log |\mathsf{S}|)$, performs the revisit step at most $O(\varepsilon^{-2} \log |\mathsf{S}|)$ times, and satisfies the $(\varepsilon, m)$-sliding-window ERM property with respect to any observation sequence $\mathcal{Z} \sim \mathcal{W}$.*

**From fixed to stable data distributions.** We note that Theorem 4.2 extends to the case where the data distribution is not fixed but *stable* under $\mathcal{W}$. Formally, for two data distributions $D$ and $D'$, the *total variation distance* is defined as $\mathrm{TV}(D, D') := \sup_{A \subseteq \mathsf{X}} |D(A) - D'(A)|$. Intuitively, $\mathrm{TV}(D, D')$ is the maximum difference in selectivity that the two distributions assign to the same point set. We say that an SD sequence $\mathcal{W}$ is $\sigma$-*stable* if for every pair $W_i, W_j \in \mathcal{W}$, $\mathrm{TV}(D_i, D_j) \leq \sigma$ holds. Theorem 4.2 remains valid as long as $\mathcal{W}$ is $c\varepsilon$-stable, where $c$ is a suitably small constant and $\varepsilon$ is the algorithm's tolerance parameter.

## 4.2 GRADUALLY DRIFTING DATA AND QUERIES

In the previous subsection, we bounded the regret when the data distribution is fixed or sufficiently stable, even if the query distribution changes arbitrarily at each step. Ideally, we would like our algorithm's prediction, based on past observations $\langle z_1, \ldots, z_t \rangle$, to remain accurate for the next observation $z_{t+1}$. Clearly, if either distribution drifts abruptly, no meaningful accuracy guarantees are possible. We therefore focus here on the case of gradual drift, adopting the drift-tracking framework of Mohri & Medina (2012).

**$(\Delta, \varepsilon)$-tracking.** Let $\mathcal{H}$ be a hypothesis set. Let ALG be an algorithm that receives a sequence $\mathcal{Z} = \langle z_1, z_2, \ldots \rangle$ of observations and maintains a hypothesis in $\mathcal{H}$. Let $h_t \in \mathcal{H}$ be the hypothesis that ALG has computed at step $t$, which depends on the prefix $\mathcal{Z}_t$ of $\mathcal{Z}$. Let $\Lambda_t = \ell_{h_{t-1}}(z_t)$ denote the loss on observation $z_t$. Let $\mathcal{W} = \langle W_1, W_2, \ldots \rangle$ be a sequence of SD's. For any $t > 0$, we define $\bar{\Lambda}_t(\mathcal{W}) = \mathbb{E}_{\mathcal{Z}_t \sim \mathcal{W}}[\Lambda_t]$. For parameters $\Delta, \varepsilon \in (0, 1)$, we say that ALG $(\Delta, \varepsilon)$-*tracks* $\mathcal{H}$ if there exists $t_0 := t_0(\Delta, \varepsilon)$ such that for all $t \geq t_0$ and for any sequence $\mathcal{W}$ where $\mathrm{disc}_{\mathcal{H}}(W_i, W_{i+1}) \leq \Delta$ for all $i \geq 1$, we have $\bar{\Lambda}(\mathcal{W}) \leq \inf_{h \in \mathcal{H}} \mathrm{err}_{W_t}(\mu) + \varepsilon$.

Intuitively, assuming that that the *drift rate* in $\mathcal{W}$ is limited to $\Delta$, a tracking algorithm is expected to deliver good predictions. In this section, we prove that DUSS is a tracking algorithm when both query and data distributions drift gradually. Before doing so, we first establish a general result that holds for any sliding-window ERM algorithm (see Section 4.1). We believe this result is of independent interest with other potential applications.

**Theorem 4.3.** *Let $\Sigma = (\mathsf{X}, \mathsf{R})$ be a range space with $\mathrm{VC\text{-}dim}(\Sigma) = O(1)$. Let $\mathcal{D}$ be a class of distributions over $\mathsf{X}$ and $\mathcal{M} := \mathcal{M}_{\Sigma, \mathcal{D}}$ the associated family of selectivity functions. Consider an $(\varepsilon, m)$-window ERM algorithm ALG using the hypothesis set $\mathcal{M}$. If the drift rate $\Delta$ of the SD sequence satisfies $\Delta = O(\varepsilon^3 \log^{-4}(\varepsilon^{-1}))$ and the window size $m = \Theta(\varepsilon^{-2} \log^4(\varepsilon^{-1}))$, then ALG $(\Delta, \varepsilon)$-tracks $\mathcal{M}$.*

Informally, the theorem suggests that for drift rate $\Delta < \varepsilon^3$, a $(\varepsilon, m)$-window ERM algorithm with window size $m \approx \varepsilon^{-2}$ can consistently maintain its prediction accuracy. The crux of the proof lies in bounding the covering number of $\mathcal{M}$. For a parameter $\alpha > 0$ and $m > 1$, the $\alpha$-*covering number*, denoted $N(\mathcal{M}, \alpha, m)$, is the smallest number of hypotheses in $\mathcal{M}$ that can approximate, within $\alpha$ point-wise error, all functions in $\mathcal{M}$ on any set of $m$ queries. We prove the following lemma. See also Lemma B.2 in Appendix B.

**Lemma 4.4.** $N(\mathcal{M}, \alpha, m) = m^{O(\alpha^{-2} \log \alpha^{-1})}$.

We combine this lemma with some known results in learning theory to obtain Theorem 4.3. See Appendix B for more details. We now apply Theorem 4.3 to DUSS. Consider a realizable sequence of SD's $\mathcal{W} = \langle W_1, W_2, \ldots \rangle$ with associated underlying data distributions $\langle D_1, D_2, \ldots \rangle$, where the total variation distance between $D_t$ and $D_{t+1}$ is at most $\Delta = c_1 \varepsilon^3 \log^{-4}(\varepsilon^{-1})$, for some constant $c_1$ to be chosen. Given a window size $m = \Theta(\varepsilon^{-2} \log^4 \varepsilon^{-1})$, we can choose $c_1$ such that for any sliding window $\mathcal{Z}_{t,m}$ and for any $W, W' \in \mathcal{W}_{t,m}$ with respective underlying data distributions $D, D'$, we have $\mathrm{TV}(D, D') \leq m\Delta \leq c_2 \varepsilon$ for some $c_2 < 1$. By the extension of Theorem 4.2 to stable data distributions, we argue that DUSS satisfies the $(\varepsilon, m)$-window ERM property. Combining this fact with Theorem 4.3, we establish the following property of DUSS, which is the main result of this section.

**Theorem 4.5.** *Let $\Sigma = (\mathsf{X}, \mathsf{R})$ be a range space with $\mathrm{VC\text{-}dim}(\Sigma) = O(1)$. Let $\mathcal{D}$ be the class of discrete distributions on $\mathsf{X}$ and $\mathcal{M} := \mathcal{M}_{\Sigma, \mathcal{D}}$ the corresponding family of selectivity functions. Let $\varepsilon \in (0, 1)$ be the error threshold, and assume that the sequence of SD's is realized by an underlying sequence of data distributions where the total-variation distance between consecutive*

*distributions is at most $\Delta = O(\varepsilon^3 \log^4(\varepsilon^{-1}))$. Using a window size $m = \Theta(\varepsilon^{-2} \log^4(\varepsilon^{-1}))$ and a $(\varepsilon/100)$-representative support,* DUSS $(\Delta, \varepsilon)$-*tracks* $\mathcal{M}$.

In simple words, the above result suggests that if the drift rate $\Delta < o(\varepsilon^3)$, we can initialize DUSS with a representative support and set its window size to $m = \Theta(\varepsilon^{-2} \log^4(\varepsilon^{-1}))$; then, DUSS maintains a model that accurately predicts selectivities of incoming queries in a way that consistently tracks the gradually drifting environment.

## 5 SUMMARY OF EXPERIMENTS

We evaluate DUSS on real-world datasets, comparing it with other query-driven methods and baselines. Since our main goal is to validate our theoretical results (instead of outperforming state-of-the-art systems), we use a simple implementation of DUSS.

**Datasets and Queries.** We use several standard real-world datasets from prior work, including **Power** (Dua & Graff, 2019), **Forest** (Dua & Graff, 2019), and **IMDb** (Leis et al., 2015). Data are normalized to $[0, 1]^d$. We further splice the **IMDb** data along the time axis to simulate data drift. For queries, we consider orthogonal range queries (boxes), since all models we compare with can support them. We consider two forms of drift. In the *gradual* drift setting, query centers shift slowly from one region of the data space to another, producing slow but continuous changes in workload. In the *abrupt* drift setting, queries remain clustered around a fixed region for some time before suddenly jumping to a different region, yielding sharp transitions. Figure D.1 illustrates the two forms of drift. Additional details regarding the datasets and query generation are deferred to Appendix D.

**Implementation details for DUSS.** Recall that DUSS assumes access to a representative support set $\mathsf{S} \subseteq [0, 1]^d$. Rather than setting it as a uniform grid over $[0, 1]^d$ or precomputing it some other way (e.g., using historical queries), we construct $\mathsf{S}$ dynamically: when queries target a region, we adaptively increase resolution there, under the assumption that future queries are likely to target nearby regions. Specifically, when a new query arrives, if fewer than MIN-PTS $= 20$ points fall inside its range, we sample additional points from within that range uniformly to ensure there are at least that many points and add them with negligible initial weights. This gives flexibility to the algorithm to tune them later if necessary. We upper-bound the model size by setting the support size budget to $K = 50,000$ points (less than 4 MB), and initialize $\mathsf{S}$ with a few thousand uniformly sampled points. When the space budget is exhausted, DUSS can compress the support set via weighted sampling, although in our experiments $\mathsf{S}$ never required compression. We also stored $\mathsf{S}$ as a simple array and performed all operations by scanning. While there are several advanced data structures that could accelerate these operations for orthogonal ranges, even this basic implementation is sufficiently efficient to validate our theory. As for the error tolerance parameter $\varepsilon$, since selectivity values are typically very small in practice (most queries return a few hundred tuples out of hundreds of thousands), we set the error tolerance parameter to $\varepsilon = 10^{-4}$. Although in theory (Theorem 4.5), the algorithm requires a sliding window of $\varepsilon^{-2}$ for tracking (no window is needed for regret guarantees per Theorem 4.2), in our experiments we observed that a much smaller window or no window worked well. Hence, for simplicity, we report results for the sliding window parameter $m = 0$.

**Methods compared.** We compare DUSS against other approaches that operate using query feedback only. Such models fall into two classes: deep learning and distribution-based. We pick one representative of each, along with a widely studied baseline: CDF (Wu et al., 2025), a recent state-of-the-art deep model; PtsHist (Hu et al., 2022), a distribution-based model; and MSCN (Kipf et al., 2019), a standard baseline. Unlike DUSS, which adapts online, none of the other methods are designed for continual updates and rely on periodic retraining or fine-tuning. To ensure fair comparison, we provide each model with $s_{\text{init}} = 2000$ initial training queries drawn from the first SD $W_1 = (Q_1, D_1)$, and then evaluate them on the test sequence $\mathcal{Z} = (z_1, z_2, \ldots)$, where each $z_t = (R_t, s_t) \sim W_t \in \mathcal{W}$. We explore three adaptation strategies: (i) $\mathfrak{M}$-$S$, where the model remains static; (ii) $\mathfrak{M}$-$R(w, p)$, where the model is retrained every $p$ queries using the most recent $w$ (or all when $w = \infty$); and (iii) $\mathfrak{M}$-$T(w, p)$, where the model is fine-tuned after every $p$ queries using $w$ recent queries (supported only by CDF and MSCN, not by PtsHist).

**Performance metrics.** We evaluate accuracy using standard Root-Mean-Squared-Error (RMSE) and percentile Q-error (Moerkotte et al., 2009). For $n$ test queries $\{R_i\}$ with estimates $\hat{s}(R_i)$ and true

Table 5.1: Selectivity estimation accuracy and training cost under simultaneous data and query distribution drifts (data drifts gradually and then abruptly; queries drift gradually or abruptly) on IMDb-7d and IMDb-2d. *The lowest error and training time in each column are highlighted in bold.*

| Method | IMDb-7d Query-Gradual | | | | IMDb-7d Query-Abrupt | | | | IMDb-2d Query-Gradual | | | | IMDb-2d Query-Abrupt | | | |
| | RMSE | *Med.* q-err | 90-th q-err | Train (s) | RMSE | *Med.* q-err | 90-th q-err | Train (s) | RMSE | *Med.* q-err | 90-th q-err | Train (s) | RMSE | *Med.* q-err | 90-th q-err | Train (s) |
|---|---|---|---|---|---|---|---|---|---|---|---|---|---|---|---|---|
| DUSS | **0.079** | **1.53** | 12.4 | 39 | **0.067** | **1.42** | **11.2** | 47 | **0.023** | 1.022 | **1.321** | 11 | **0.013** | **1.005** | **1.081** | 11 |
| CDF-$R$ (2k, 2k) | 0.287 | 2.16 | 12.2 | 733 | 0.341 | 3.89 | 299.0 | 748 | 0.212 | 1.122 | 2.483 | 121 | 0.452 | 1.484 | 12.510 | 60 |
| CDF-$R$ ($\infty$, 2k) | 0.215 | 1.64 | **7.4** | 2000 | 0.209 | 1.88 | 86.0 | 2867 | 0.099 | 1.050 | 1.879 | 260 | 0.335 | 1.201 | 113.900 | 255 |
| MSCN-$R$ (2k, 2k) | 0.285 | 2.35 | 12.8 | 120 | 0.324 | 3.25 | 49.1 | 128 | 0.189 | 1.174 | 1.792 | 21 | 0.431 | 1.358 | 11.540 | 13 |
| MSCN-$R$ ($\infty$, 2k) | 0.229 | 1.76 | 7.4 | 382 | 0.266 | 2.25 | 29.5 | 381 | 0.149 | 1.034 | 1.378 | 54 | 0.358 | 1.036 | 16.750 | 48 |
| PtsHist-$R$ (2k, 2k) | 0.110 | 2.93 | 284.0 | 505 | 0.116 | 4.38 | 594.0 | 512 | 0.029 | **1.011** | 1.520 | 459 | 0.035 | 1.006 | 1.154 | 439 |
| PtsHist-$R$ ($\infty$, 2k) | 0.106 | 2.57 | 212.1 | 1280 | 0.106 | 3.33 | 306.0 | 1398 | 0.027 | 1.014 | 1.606 | 4553 | 0.030 | 1.007 | 1.166 | 4839 |

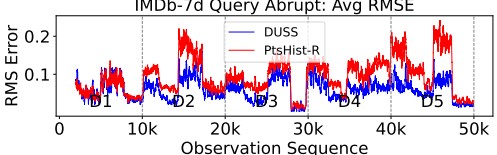 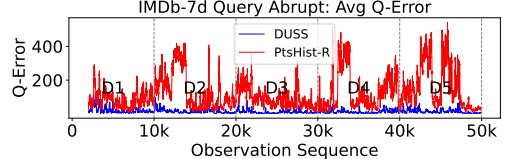

Figure 5.1: Sliding-window performance on IMDb-7d.

selectivities $s(R_i)$, RMSE is $\left(\frac{1}{n}\sum_{i=1}^{n}(\hat{s}(R_i) - s(R_i))^2\right)^{1/2}$, and Q-error$(p)$ is the $p$-th percentile of $\{\max\{\hat{s}(R_i), s(R_i)\}/\min\{\hat{s}(R_i), s(R_i)\}\}$. RMSE captures absolute error, while Q-error highlights relative error and is widely used in the database community since selectivities are often small. We also report efficiency, measure training/fine-tuning overhead and inference time in Appendix D.

**Summary.** We perform two types of experiments: (i) fixing the data distribution while allowing the query distribution to drift (more details in Appendix D.1), and (ii) allowing both data and query distributions to drift (more details in Appendix D.2). Within each setting, we consider both gradual and abrupt drift for queries and, where applicable, for data. Across all scenarios, DUSS consistently delivers the best trade-off between accuracy and efficiency, while also incurring some of the lowest training (model-update) costs.

Under gradual query drift, the distribution-based PtsHist can achieve comparable, and occasionally marginally better, predictive performance on certain metrics in low-dimensional cases. This observation aligns with our theory (Theorem 4.5), which suggests that distribution-based models, by maintaining a good fit on a sliding window, should also perform well under gradual drift. However, PtsHist requires solving a quadratic program, leading to significantly higher training (model-update) time. In contrast, the neural network–based CDF-MSCN and MSCN are considerably less effective, even with frequent retraining. We suspect that their model complexity demands much larger training data and longer sliding windows to generalize effectively. See Table 5.1 here and Table D.1 in appendix for more details. Unfortunately, PtsHist loses its advantage in high dimensions: because it is designed around a fixed support set, it cannot maintain a representative support in sparse, high-dimensional spaces. Preserving its performance would require dramatically increasing the support size. In contrast, DUSS adapts through a dynamic support.

When drift becomes abrupt—especially in high-dimensional settings—DUSS is the only method that maintains superior accuracy with minimal training costs, owing to its low-regret guarantees (Theorem 4.2). Again, see Table 5.1 and Table D.1 for details. Figure 5.1 further illustrates a complex drift scenario with abrupt query drift and mixed data drift (further details are in Appendix D.2). DUSS consistently maintains lower average sliding-window error, and quickly adapts under drift, demonstrating its robustness.

Taken together, these results these results confirm that DUSS can maintain robust estimator even in the presence of drift by only observing query-selectivity (query-cardinality) pairs.

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

---

**Algorithm 1** DUSS: Dynamic Update Support-Set.

---

1: **Input:** Window size $m$, error parameter $\varepsilon$, support set $\mathsf{S} \subseteq \mathsf{X}$
2: Initialize weights: $\omega(p) \leftarrow 1$ for all $p \in \mathsf{S}$
3: $W_{\text{curr}} \leftarrow |\mathsf{S}|, W_{\text{rev}} \leftarrow |\mathsf{S}|, \text{COUNT} \leftarrow 0$
4: Initialize $\widehat{D}$ as the distribution with uniform weights on $\mathsf{S}$
5: **Method** PROCESS(Observation stream $\mathcal{Z}$)
6:   **loop**   $z_t = (R_t, s_t) \leftarrow$ each new observation
7:     $\mathcal{Z}_{t,m} \leftarrow (\mathcal{Z}_{t-1,m} \cup \{z_t\}) - \{z_{t-m}\}, \quad \hat{s}_t \leftarrow \mu_{\widehat{D}}(R_t)$
8:     **if** $|\hat{s}_t - s_t| > \varepsilon$ **then**
9:       WEIGHTUPDATE$((R_t, s_t))$
10:       **while** $W_{\text{curr}} > W_{\text{rev}}/(1 - \varepsilon/2)$ **do**
11:         $W_{\text{rev}} \leftarrow W_{\text{curr}}$
12:         REVISITWINDOW
13: **Method** WEIGHTUPDATE$((R, s))$
14:   **if** $\mu_{\widehat{D}}(R) < s - \varepsilon$ **then**
15:     $\chi \leftarrow \frac{\varepsilon^2}{4(s-\varepsilon/2)}$
16:     **while** $\mu_{\widehat{D}}(R) < s - \varepsilon$ **do**
17:       $\omega(p) \leftarrow (1 + \chi) \cdot \omega(p), \forall p \in \mathsf{S} \cap R$
18:       $\text{COUNT} \leftarrow \text{COUNT} + 1$
19:   **else if** $\mu_{\widehat{D}}(R) > s + \varepsilon$ **then**
20:     $\chi \leftarrow \frac{\varepsilon^2}{4(1-s-\varepsilon/2)}$
21:     **while** $\mu_{\widehat{D}}(R) > s + \varepsilon$ **do**
22:       $\omega(p) \leftarrow (1 + \chi) \cdot \omega(p), \forall p \in \mathsf{S} \setminus R$
23:       $\text{COUNT} \leftarrow \text{COUNT} + 1$
24:   **if** $\text{COUNT} > \tau_{\text{res}}$ **then**
25:     RESET
26:     **return**

---

Peizhi Wu and Zachary G. Ives. Modeling shifting workloads for learned database systems. *Proc. ACM Manag. Data*, 2(1):38:1–38:27, 2024. doi: 10.1145/3639293. URL https://doi.org/10.1145/3639293.

Peizhi Wu, Haoshu Xu, Ryan Marcus, and Zachary G Ives. A practical theory of generalization in selectivity learning. *Proceedings of the VLDB Endowment*, 18(6), 2025.

Haibo Xiu, Pankaj K. Agarwal, and Jun Yang. PARQO: penalty-aware robust plan selection in query optimization. *Proc. VLDB Endow.*, 17(13):4627–4640, 2024. URL https://www.vldb.org/pvldb/vol17/p4627-xiu.pdf.

Zongheng Yang, Eric Liang, Amog Kamsetty, Chenggang Wu, Yan Duan, Xi Chen, Pieter Abbeel, Joseph M Hellerstein, Sanjay Krishnan, and Ion Stoica. Deep unsupervised cardinality estimation. *Proceedings of the VLDB Endowment*, 13(3), 2020.

Sepanta Zeighami and Cyrus Shahabi. Towards establishing guaranteed error for learned database operations. In *The Twelfth International Conference on Learning Representations, ICLR 2024, Vienna, Austria, May 7-11, 2024*. OpenReview.net, 2024a. URL https://openreview.net/forum?id=6tqgL8VluV.

Sepanta Zeighami and Cyrus Shahabi. Theoretical analysis of learned database operations under distribution shift through distribution learnability. In *Forty-first International Conference on Machine Learning, ICML 2024, Vienna, Austria, July 21-27, 2024*. OpenReview.net, 2024b. URL https://openreview.net/forum?id=oowQ8LPA12.

## A  PROOF OF LEMMA 4.1

For a range $R$, let $\bar{R}$ denote its complement. If $R$ is a box then $\bar{R}$ is the region lying outside the box. For $i \geq 1$, let $z_i = (R_i, s_i)$ be the observation when the weights were updated the $i$-th time, let $W_i$

be the value of $W_{\text{curr}}$ after $i$ weight updates. Recall, that an observation may cause multiple updates, so $z_i$ may be the same as $z_{i+1}$.

Recall that the support $S$ is $c\varepsilon$-representative. For the analysis, assume $c = 0.01$. Let $A \subseteq S$ be an $c\varepsilon$-sample of $D^*$. Since the input data distribution $D^*$ is fixed, by the $c\varepsilon$-representative support property of $S$, such a set must exist. We prove a bound on $T$, the number of weight-update steps, by obtaining an upper bound on $W_T$ and a lower bound on the weight of the points in $A$ after $T$ updates, and by showing that the latter exceeds the former once $T > \tau_{\text{res}}(\varepsilon)$. This is a typical argument for the multiplicative-weight-update (MWU) method Arora et al. (2012).

We first obtain an upper bound on $W_i$. Initially, $W_0 = |S|$. For simplicity, let $\gamma = \varepsilon/2$. If $R_i$ is light, we set $b_i = s_i - \gamma$ and $\Delta_i = R_i$, and if $R_i$ is heavy, we set $b_i = 1 - s_i - \gamma$ and $\Delta_i = \bar{R}_i$. Then after the $i$-th weight update,

$$
\begin{aligned}
W_i &\leq \sum_{p \in \Delta_i} \omega(p)\left(1 + \frac{\gamma^2}{b_i}\right) + \sum_{p \notin \Delta_i} \omega(p) \\
&\leq W_{i-1} + W_{i-1} \cdot \frac{\gamma^2}{b_i} \sum_{p \in \Delta_i} \frac{\omega(p)}{W_{i-1}} \leq W_{i-1}\left(1 + \frac{\gamma^2}{b_i} \cdot (b_i - \gamma)\right) \\
&\leq W_{i-1}(1 + \gamma^2(1 - \gamma)) \leq W_{i-1} \cdot \exp\big(\gamma^2(1 - \gamma)\big).
\end{aligned}
$$

The second last inequality follows because $b_i \leq 1$. Hence,

$$
W_T \leq W_0 \cdot \exp\big(\gamma^2(1 - \gamma)T\big) = |S| \cdot \exp\big(\gamma^2(1 - \gamma)T\big).
$$

Next, we focus on the weights of points in $A$. First, it can be verified that $b_i \geq \gamma$. Let $W_i(p)$ denote the weight of $p \in S$ after the $i$-th weight update. If $p \in \Delta_i$, then

$$
W_i(p) \geq (1 + \frac{\gamma^2}{b_i})W_{i-1}(p) \geq (1 + \gamma)^{\frac{\gamma}{b_i}} W_{i-1}(p). \tag{2}
$$

The last inequality follows because $\gamma, \gamma/b_i \in [0, 1]$. For a subset $I \subseteq [T]$, let $\sigma(I) = \sum_{j \in I} (1/b_j)$. Let $\mathcal{I}(p) \subseteq \{1, \dots, T\}$ be the set of indices in which the weigh of $p$ was updated, then by (2),

$$
W_T(p) \geq W_0(p) \cdot (1 + \gamma)^{\gamma \cdot \sigma(\mathcal{I}(p))}. \tag{3}
$$

Since arithmetic mean of a set of non-negative numbers is at least as large as their geometric mean

$$
\Big[\prod_{p \in A} W_T(p)\Big]^{\frac{1}{a}} \leq \frac{W_T(A)}{a} \leq \frac{W_T}{a}, \tag{4}
$$

where $a = |A|$ and $W_T(A) = \sum_{p \in A} W_T(p)$. On the other hand, by (3)

$$
\Big[\prod_{p \in A} W_T(p)\Big]^{\frac{1}{a}} \geq (1 + \gamma)^{\frac{\gamma}{a} \cdot \sum_{p \in A} \sigma(\mathcal{I}(p))}. \tag{5}
$$

Next, we observe that

$$
\frac{1}{a} \sum_{p \in A} \sigma(\mathcal{I}(p)) = \frac{1}{a} \sum_{p \in A} \sum_{j \in \mathcal{I}(p)} \frac{1}{b_j} = \frac{1}{a} \sum_{j=1}^{T} \frac{|A \cap \Delta_j|}{b_j}. \tag{6}
$$

Since $\mathcal{A}$ is an $c\varepsilon$-sample, for $c = 0.01$, of the underlying data distribution $D^*$, $\frac{|A \cap \Delta_j|}{a} \geq b_j + 0.98\gamma$. Plugging this bound in (6),

$$
\frac{1}{a} \sum_{p \in A} \sigma\big(\mathcal{I}(p)\big) \geq \sum_{j=1}^{T} \frac{b_j + 0.98\gamma}{b_j} \geq T \cdot (1 + 0.98\gamma). \tag{7}
$$

Combining (2),(3), and (7), we obtain

$$(1 + \gamma)^{\gamma(1+0.98\gamma)T} \leq \frac{W_T}{a} \leq W_T. \tag{8}$$

Plugging the value of $W_T$ in (7) and taking $\ln$ on both sides,

$$\gamma T (1 + 0.98\gamma) \ln(1 + \gamma) \leq \gamma^2 (1 - \varepsilon)T + \ln|\mathsf{S}|.$$

Using the fact that $\ln(1 + \gamma) \geq \gamma - \gamma^2/2$, we obtain $T \leq \gamma^{-2}\ln|\mathsf{S}|(f(\gamma))^{-1}$, where $f(\gamma) = (1 + 0.98\gamma) \cdot (1 - \gamma/2) - (1 - \gamma) \geq \gamma/2$. Hence, substituting $\gamma = \varepsilon/2$, we conclude that $T \leq 2\gamma^{-3} \ln|\mathsf{S}| \leq 16\varepsilon^{-3} \ln|\mathsf{S}|$, as claimed. This completes the proof of the lemma.

## B    SLIDING-WINDOW SIZE BOUND UNDER THE DYNAMIC SETTING

**Theorem B.1** (Restatement of Theorem 4.3). *Let $\Sigma = (\mathsf{X}, \mathsf{R})$ be a range space with* VC-dim$(\Sigma) = O(1)$. *Let $\mathcal{D}$ be a class of distributions over $\mathsf{X}$ and $\mathcal{M} := \mathcal{M}_{\Sigma,\mathcal{D}}$ the associated family of selectivity functions. Consider an $(\varepsilon, m)$-window ERM algorithm* ALG *using the hypothesis set $\mathcal{M}$. If the drift rate $\Delta$ of the SD sequence satisfies $\Delta = O(\varepsilon^3 \log^{-4}(\varepsilon^{-1}))$ and the window size $m = \Theta(\varepsilon^{-2} \log^4(\varepsilon^{-1}))$, then* ALG $(\Delta, \varepsilon)$-*tracks $\mathcal{M}$.*

The argument proceeds in several parts. The key insight, however, is to show that the $\alpha$-*cover* (see (Shalev-Shwartz & Ben-David, 2014, Ch. 26–27)) of the class of selectivity functions $\mathcal{M}$ is small. We then use the bound on the $\alpha$-cover to bound the *Rademacher complexity*, a well-known concept in machine learning (Shalev-Shwartz & Ben-David, 2014, Chapter 26). Finally, we combine the bound on the Rademacher complexity with a result by (Mohri & Medina, 2012, Theorem 1) to prove Theorem 4.3.

Let $\mathcal{B} \subseteq \mathsf{R}$ be a set of ranges. For two selectivity functions $\mu_1, \mu_2 \in \mathcal{M}$, we define the distance between them with respect to $\mathcal{B}$ to be $d_{\mathcal{B}}(\mu_1, \mu_2) := \max_{R \in \mathcal{B}} |\mu_1(R) - \mu_2(R)|$. For $\alpha > 0$, a subset $\mathcal{M}' \subseteq \mathcal{M}$ is called an $\alpha$-*cover* with respect to $\mathcal{B}$ if all functions of $\mathcal{M}$ are within distance $\alpha$ from $\mathcal{M}'$ (under the distance function $d_{\mathcal{B}}$). That is, $\sup_{\mu \in \mathcal{M}} \inf_{\mu \in \mathcal{M}'} d_{\mathcal{B}}(\mu, \mu') \leq \alpha$. We define the *empirical $\alpha$-covering number* (w.r.t. $\mathcal{B}$) of $\mathcal{M}$ as

$$N(\mathcal{M}, \alpha, \mathcal{B}) = \min\{|\mathcal{M}'| : \mathcal{M}' \text{ is an } \alpha\text{-cover of } \mu \text{ w.r.t. } \mathcal{B}\}.$$

Finally, for $m \geq 1$, set $N(\mathcal{M}, \alpha, m) = \max_{\mathcal{B} \subseteq \mathsf{R}, |\mathcal{B}|=m} N(\mathcal{M}, \alpha, \mathcal{B})$.

Our main technical result is an upper bound on $N(\mathcal{M}, \alpha, m)$ stated below.

**Lemma B.2.** *Let $\Sigma = (\mathsf{X}, \mathsf{R})$ be a range space with finite VC-dimension. Let $\mathcal{D}$ be a class of probability distributions over $\mathsf{X}$ and let $\mathcal{M} := \mathcal{M}_{\Sigma,\mathcal{D}}$. For any $\alpha > 0$ and positive integer $m$,*

$$N(\mathcal{M}, \alpha, m) = m^{O(\alpha^{-2} \log \alpha^{-1})}.$$

*Proof.* Let $\mathcal{B} \subseteq \mathsf{R}$ be any arbitrary subset of $m$ ranges. We bound $N(\mathcal{M}, \alpha, \mathcal{B})$ in three steps. First, we show that there exists a family $\mathcal{C}$ of uniform discrete distributions each with support size $\eta = O(\alpha^{-2} \log \alpha^{-1})$, such that $\mathcal{M}_{\mathcal{C}} = \{\mu_C \mid C \in \mathcal{C}\}$, the class of selectivity functions associated with the distributions in $\mathcal{C}$, forms an $\alpha/2$-cover of $\mathcal{M}$ with respect to $\mathcal{B}$ (Note that even if $\mathcal{M}_{\mathcal{C}} \nsubseteq \mathcal{M}$, the notion of $\alpha/2$-cover is still well-defined). Next, we show that there exists a small subset $\mathcal{C}' \subseteq \mathcal{C}$ such that $\mathcal{M}_{\mathcal{C}} = \mathcal{M}_{\mathcal{C}'}$. Finally, we use $\mathcal{C}'$ to choose a subset $\mathcal{D}' \subseteq \mathcal{D}$ of size $|\mathcal{C}'|$ and set $\mathcal{M}' = \{\mu_D \mid D \in \mathcal{D}'\}$ such that $\mathcal{M}'$ is an $\alpha$-cover of $\mathcal{M}$ (w.r.t. $\mathcal{B}$). We describe the full construction below.

Let $\mathcal{B} \subseteq \mathsf{R}$ be any arbitrary subset of $m$ ranges. Consider the range space $\Sigma_{\mathcal{B}} = (\mathsf{X}, \mathcal{B})$. It is easily seen that $\mathrm{VC}(\Sigma_{\mathcal{B}}) \leq \mathrm{VC}(\Sigma)$, so $\mathrm{VC}(\Sigma_{\mathcal{B}}) = O(1)$. For any distribution $D_i \in \mathcal{D}$, as mentioned above, there is an $(\alpha/2)$-sample $C_i$ of size $\eta = O(\alpha^{-1} \log \alpha^{-1})$, i.e., $|\mu_{D_i}(R) - \frac{|C_i \cap R|}{|C_i|}| \leq \alpha/2$ for any $R \in \mathsf{R}$. Setting $\mu_{C_i}(R) = \frac{|C_i \cap R|}{|C_i|}$, $d_{\mathcal{B}}(\mu_{C_i}, \mu_{D_i}) \leq \alpha/2$. Let $\mathcal{C} = \{C_i \mid D_i \in \mathcal{D}\}$. Then $\mathcal{M}_{\mathcal{C}}$ is an $(\alpha/2)$-cover of $\mathcal{M}$ (w.r.t. $\mathcal{B}$).

Next, we choose the set $\mathcal{C}' \subseteq \mathcal{C}$ as follows. Define the dual range space $\Sigma_{\mathcal{B}}^* = (\mathcal{B}, \mathsf{X}^*)$ of $\Sigma_{\mathcal{B}}$ where
$$\mathsf{X}^* = \{\{R \in \mathcal{B} \mid x \in R\} \mid x \in \mathsf{X}\}.$$

Each range in $\mathsf{X}^*$ is defined by a point $x \in \mathsf{X}$ and comprises the set of ranges in $\mathcal{B}$ that contain $x$. Since VC-dim$(\Sigma_{\mathcal{B}}) = O(1)$ then VC-dim$(\Sigma_{\mathcal{B}}^*)$is also $O(1)$, say VC-dim$(\Sigma^*) = \kappa$. Then $|\mathsf{X}^*| = O(m^\kappa)$. $\mathsf{X}^*$ implies an equivalence relation $\equiv$ on $\mathsf{X}$ that partitions $\mathsf{X}$ into $O(m^\kappa)$ equivalence classes, where $x_1 \equiv x_2$ if and only if any range in $\mathcal{B}$ either contains both $x_1$ and $x_2$ or neither of them Chazelle & Welzl (1989). Define two subsets $C, C' \subseteq \mathsf{X}$ as equivalent with respect to $\mathcal{B}$ if there exists a bijection $f : C \to C'$ such that $x \equiv f(x)$ for all $x \in \mathsf{X}$. The number of combinatorially distinct subsets (with respect to $\mathcal{B}$) of $\mathsf{X}$ of size $\eta$ is at most $O(m^{\eta \cdot \kappa})$. Observe that if $C \equiv C'$, then $\mu_C(R) = \mu_{C'}(R)$ for every $R \in \mathcal{B}$. Let $\mathcal{C}' \subseteq \mathcal{C}$ be a maximal set of combinatorially distinct sets (i.e. they are defined by combinatorially distinct subsets of $\mathsf{X}$) in $\mathcal{C}$. Since $|C| \leq \eta$ for any $C \in \mathcal{C}$, $|\mathcal{C}| = O(m^{\kappa \cdot \eta})$. Finally. we choose a subset $\mathcal{D}' \subseteq \mathcal{D}$ of size $|\mathcal{C}'|$ as follows. Recall that each $C_i \in \mathcal{C}$ is an $(\alpha/2)$-sample pf a distribution $D_i \in \mathcal{D}$. Set $\mathcal{D}' = \{D_i \mid C_i \in \mathcal{C}'\}$ and $\mathcal{M}' = \{\mu_D \mid D \in \mathcal{D}'\}$. To prove that $\mathcal{M}'$ is an $\alpha$-cover of $\mathcal{M}$, let $D_i$ be a distribution in $D$, let $C_i' \in \mathcal{C}$ be the set equivalent to $C_i$, and let $D_i' \in \mathcal{D}'$ be the distribution in $\mathcal{D}'$ corresponding to $\mathcal{C}_i'$. Then for any $R \in \mathcal{B}$, $|\mu_{D_i}(R) - \mu_{D_i'}(R)| \leq \alpha/2 + \alpha/2 \leq \alpha$ (using the triangle inequality). Hence, $d_{\mathcal{B}}(\mu_{D_i}, \mu_{D_i'}) \leq \alpha$, so $\mathcal{M}'$ is an $\alpha$-cover of $\mathcal{M}$ of size $O(m^{\kappa \cdot \eta}) = m^{O(\alpha^{-2} \log \alpha^{-1})}$.

$\square$

To proceed with the proof we require the following definition.

**Rademacher complexity.** Let $\mathcal{B} = \{R_1, R_2, \ldots, R_m\} \subseteq \mathsf{R}$, be a subset of $m$ ranges. Let $\sigma = (\sigma_1, \ldots, \sigma_m) \in \{+1, -1\}^m$ be a random vector where $\Pr[\sigma_i = 1] = \Pr[\sigma_i = 0] = 1/2$. The *empirical Rademacher complexity* of $\mathcal{M}$ w.r.t. $\mathcal{B}$ is defined as

$$\hat{\mathfrak{R}}_{\mathcal{B}}(\mathcal{M}) := \frac{1}{m} \mathbb{E}_\sigma \left[ \sup_{\mu \in \mathcal{M}} \sum_{i=1}^{m} \sigma_i \, \mu(R_i) \right].$$

For $m \geq 1$, we define the (worst-case) empirical Rademacher complexity as $\hat{\mathfrak{R}}_m(\mathcal{M}) := \sup_{\mathcal{B} \subseteq \mathsf{R}, |\mathcal{B}| = m} \hat{\mathfrak{R}}_{\mathcal{B}}(\mathcal{M})$.

Roughly speaking, Rademacher complexity measures the rate of uniform convergence as a function of training sample size. Using Lemma B.2 and the well-known connection between the covering number and the Rademacher complexity (see (Shalev-Shwartz & Ben-David, 2014, Ch 27)) we obtain the following lemma.

**Lemma B.3.** *Let $\Sigma = (\mathsf{X}, \mathsf{R})$ be a range space with $\mathrm{VC}(\Sigma) = O(1)$. Let $\mathcal{D}$ be a family of distributions on $\mathsf{X}$, and let $\mathcal{M} := \mathcal{M}_{\Sigma, \mathcal{D}}$. Then for any $m \geq 1$, $\hat{\mathfrak{R}}_m(\mathcal{M}) = O(m^{-1/2} \log^2 m)$.*

Recall the loss function $\ell_\mu : \mathsf{Z} \mapsto [0, 1]$ defined with respect to a selectivity function $\mu$ in Section 2. Consider the class of functions $\mathcal{L}_{\mathcal{M}} = \{\ell_\mu : \mu \in \mathcal{M}\}$. The notion of Rademacher complexity also applies to the function class $\mathcal{L}_{\mathcal{M}}$, by substituting $\mathsf{R}$ with $\mathsf{Z}$ and replacing $\mathcal{B}$ with a subset of $\mathsf{Z}$ in the definition above. Since $\mathcal{M} \subseteq \{\mathsf{R} \mapsto [0, 1]\}$, it is known that, see (Shalev-Shwartz & Ben-David, 2014, Chapter 26), $\hat{\mathfrak{R}}_m(\mathcal{L}_{\mathcal{M}}) = O(\hat{\mathfrak{R}}_m(\mathcal{M}))$. Therefore,

**Corollary B.1.** *For any $m \geq 1$, $\hat{\mathfrak{R}}_m(\mathcal{L}_{\mathcal{M}}) = O(m^{-1/2} \log^2 m)$.*

We next prove Theorem 4.3 using the Corollary B.1. By plugging the bound on $\hat{\mathfrak{R}}_m(\mathcal{L}_{\mathcal{M}}) = O(m^{-1/2} \log^2 m)$ from Corollary B.1, into a result by Mohri and Medina (Mohri & Medina, 2012, Theorem 1), we obtain the following:

**Lemma B.4.** *Let $\Sigma = (\mathsf{X}, \mathsf{R})$ be a range space with $\mathrm{VC\text{-}dim}(\Sigma) = O(1)$, and let $W_1, \ldots, W_k$ be SD's on $\mathsf{Z} = \mathsf{R} \times [0, 1]$. Suppose that $z_1, \ldots, z_k$ are observations with each $z_i \sim W_i$. Then, for any SD $W$, for any $\delta \in (0, 1)$, the following inequality holds for every $\mu \in \mathcal{M}$ with probability at least $1 - \delta$:*

$$\mathrm{err}_W(\mu) \leq \frac{1}{k} \sum_{i=1}^{k} \left( \ell_\mu(z_i) + \mathrm{disc}_{\mathcal{M}}(W_i, W) \right) + \frac{O(\log^2 k + \sqrt{\log \delta^{-1}})}{\sqrt{k}}. \tag{9}$$

*Moreover, if $\mu^* = \arg\min_{\mu \in \mathcal{M}} \mathrm{err}_W(\mu)$ then with probability at least $1 - \delta$ it holds that,*

$$\frac{1}{k} \sum_{i=1}^{k} \ell_{\mu^*}(z_i) \leq \mathrm{err}_W(\mu^*) + \frac{1}{k} \sum_{i=1}^{k} \mathrm{disc}_{\mathcal{M}}(W_i, W) + \frac{O(\log^2 k + \sqrt{\log \delta^{-1}})}{\sqrt{k}}. \tag{10}$$

We use Lemma B.4 to prove Theorem 4.3 as follows. Fix a window size $m$ and define ALG as a $(\varepsilon, m)$-sliding-window ERM algorithm that receives a sequence $\mathcal{Z} = \langle z_1, z_2, \ldots \rangle$ of observations and for each $t$ maintains a selectivity function $\mu_t$ such that

$$\frac{1}{m} \sum_{i=t-m+1}^{t} \ell_{\mu_t}(z_i) \leq \varepsilon + \inf_{\mu \in \mathcal{M}} \frac{1}{m} \sum_{i=t-m+1}^{t} \ell_{\mu_t}(z_i)$$

Suppose we execute ALG on a stream of observations $\mathcal{Z} = \langle z_1, z_2, \ldots \rangle$ drawn from a sequence $\mathcal{W} = \langle W_1, W_2, \ldots \rangle$ such that, $\mathrm{disc}_{\mathcal{M}}(\mu_i, \mu_{i+1}) \leq \Delta$ for all $i$, where $\Delta \geq 0$. Let $\mu_t^* = \arg\inf_{\mu \in \mathcal{M}} \mathrm{err}_{W_t}(\mu)$. Consider the random variable $X_{t+1}$:

$$X_{t+1} := \mathrm{err}_{W_{t+1}}(\mu_t) - \mathrm{err}_{W_{t+1}}(\mu_{t+1}^*).$$

Our goal is to bound $\mathbb{E}_{\mathcal{Z} \sim \mathcal{W}}[X_{t+1}]$. By Fubini's theorem,

$$\mathbb{E}_{\mathcal{Z} \sim \mathcal{W}}[X_{t+1}] = \mathbb{E}_{\mathcal{Z}_t \sim \mathcal{W}}[\mathrm{err}_{W_{t+1}}(\mu_t) - \mathrm{err}_{W_{t+1}}(\mu_{t+1}^*)].$$

Since ALG is an $(\varepsilon, m)$-sliding window ERM, we invoke Lemma B.4 on both $\mu_t$ and $\mu_{t+1}^*$ with a confidence parameter of $\delta/2$. By the union bound, it follows that both bounds hold simultaneously with probability at least $1 - \delta$. Combining (9) and (10) it follows that with probability $1 - \delta$,

$$X_{t+1} \leq \varepsilon + \frac{2}{m} \sum_{i=t-m+1}^{t} \mathrm{disc}_{\mathcal{M}}(\mu_i, \mu_{t+1}) + 2 \cdot \frac{O(\log^2 m + \sqrt{\log \delta^{-1}})}{\sqrt{m}}.$$

It is easy to verify that for any SD's $W_1, W_2, W_3$ over Z, the triangle inequality holds, i.e., $\mathrm{disc}_{\mathcal{M}}(W_1, W_3) \leq \mathrm{disc}_{\mathcal{M}}(W_1, W_2) + \mathrm{disc}_{\mathcal{M}}(W_2, W_3)$ holds. Since discrepancy satisfies the triangle inequality and $\mathrm{disc}_{\mathcal{M}}(W_j, W_{j+1}) \leq \Delta$ for all $j$, $\mathrm{disc}_{\mathcal{M}}(\mu_i, \mu_t) \leq (t - i)\Delta$. Thus,

$$\frac{1}{m} \sum_{i=t-m+1}^{t} \mathrm{disc}_{\mathcal{M}}(\mu_i, \mu_t) \leq (m + 1)\Delta.$$

Therefore, with probability at least $1 - \delta$,

$$X_{t+1} \leq \varepsilon + (m + 1) \cdot \Delta + 2 \cdot \frac{O(\log^2 m + \sqrt{\log \delta^{-1}})}{\sqrt{m}}.$$

We wish to bound the expectation $\mathbb{E}[X_{t+1}]$. Let $Y$ be a random variable such that $\Pr[Y > \sqrt{\log \delta^{-1}}] < \delta$ or for any integer $a \geq 1$ $\Pr[Y > a] \leq 2^{-a^2}$, we obtain that $\mathbb{E}[Y] = O(1)$. Hence, we conclude that $\mathbb{E}[X_{t+1}] \leq \varepsilon + (m + 1)\Delta + O(m^{-1/2} \log^2 m)$. This immediately implies the following lemma.

**Lemma B.5.** *Let $\mathcal{A}$ be a $(\varepsilon, m)$-sliding-window ERM algorithm (as described above). Then for any SD sequence $\mathcal{W} = \langle W_1, W_2, \ldots, \rangle$,*

$$\mathrm{err}_{W_{t+1}}(\mu_t) \leq \varepsilon + \inf_{\mu \in \mathcal{M}} \mathrm{err}_{W_{t+1}}(\mu) + O(m^{-1/2} \log^2 m) + (m + 1)\Delta.$$

In the above lemma, the second term is the estimation error (arising from the Rademacher complexity) while the last term reflects the cumulative drift over a window of size $m$. To minimize the prediction error we balance these two terms and set $m = \Theta(\Delta^{-2/3} \log^{4/3} \Delta^{-1})$. This back yields an overall excess error of $O(\Delta^{1/3} \log^{4/3} \Delta^{-1})$. Therefore, to achieve an error of $\varepsilon$, one must have $\Delta = O(\varepsilon^3 / \log^4(\varepsilon^{-1}))$. This completes the proof of Theorem 4.3.

## C  SAMPLE COMPLEXITY BOUND UNDER THE STATIC SETTING

We first state the definition of $(\varepsilon, \delta)$-learnability Haussler (1992).

$(\varepsilon, \delta)$-**learnability.** A *learning procedure* is a function ALG from a finite sequence of observations from Z to a hypothesis in $\mathcal{H}$. Namely, given a finite sequence $z^n = (z_1, \ldots, z_n) \in \mathsf{Z}^n$, ALG$(z^n)$ returns a function in $\mathcal{H}$.

We say that ALG $(\varepsilon, \delta)$-*learns* from $n$ random samples $z^n$ if

$$\sup_W \Pr\left[\mathrm{err}_W\left(\mathrm{ALG}(z^n)\right) > \inf_{\mu \in \mathcal{M}} \mathrm{err}_W(\mu) + \varepsilon\right] \leq \delta. \tag{11}$$

A hypothesis set $\mathcal{H}$ is said to be $(\varepsilon, \delta)$-*learnable* if there exists a learning procedure ALG, such that for every $\varepsilon > 0$ and $\delta > 0$, there is s sample size $n_0 = n_0(\varepsilon, \delta)$ such that (11) holds. It is called $\varepsilon$-*learnable* if the above holds for every $\varepsilon > 0$ with some fixed confidence parameter $\delta < 1$.

Next, recall the definition of $\mathcal{L}_{\mathcal{M}}$ from Appendix B. In Corollary B.1, we proved that for any $m \geq 1$, $\hat{\mathfrak{R}}_m(\mathcal{L}_{\mathcal{M}}) = O(m^{-1/2} \log^2 m)$. A well-known relationship between the Rademacher complexity and sample complexity (Shalev-Shwartz & Ben-David, 2014, Theorem 26.5) implies the following theorem

**Theorem C.1.** *Let $\Sigma = (\mathsf{X}, \mathsf{R})$ be a range space such that* VC-dim$(\Sigma) = O(1)$. *Let $\mathcal{D}$ be a family of probability distribution over $\mathsf{X}$. Then, $\mathcal{M} := \mathcal{M}_{\Sigma, \mathcal{D}}$ is agnostic $(\varepsilon, \delta)$-learnable with sample complexity $O(\varepsilon^{-2}(\log^4 \varepsilon^{-1} + \log \delta^{-1}))$ for any $\varepsilon, \delta \in (0, 1)$.*

Suppose VC-dim$(\Sigma) = \kappa_1$ and VC-dim$(\Sigma^*) = \kappa_2$, where $\Sigma^*$ corresponds to the dual range space of $\Sigma$. The order notation in the sample complexity bound of Theorem C.1 hides a dependence on $\kappa_1 \cdot \kappa_2$. It is a well-known fact that for standard geometric ranges such as boxes, balls and halfspaces in $\mathbb{R}^d$, $\kappa_1 \cdot \kappa_2 = O(d^2)$.

**Corollary C.1** (restatement of Theorem 2.1). *Let $\Sigma = (\mathsf{X}, \mathsf{R})$ be a range space, where $\mathsf{X} \subseteq \mathbb{R}^d$ and let $\mathsf{R}$ correspond to standard geometric ranges such as boxes, balls or halfspaces. Let $\mathcal{D}$ be a family of probability distribution over $\mathsf{X}$ and let $\mathcal{M} := \mathcal{M}_{\Sigma, \mathcal{D}}$ be the corresponding family of selectivity functions. Then, $\mathcal{M}$ is $\varepsilon$-learnable with sample complexity $O(d^2 \varepsilon^{-2}(\log^4 \varepsilon^{-1}))$ for any $\varepsilon \in (0, 1)$*

## D  DETAILED EXPERIMENTAL RESULTS

**Datasets.** We use several real-world datasets, all of which have used in prior work, including the benchmark study Wang et al. (2021):

- **Power** Dua & Graff (2019): Electric power usage data collected from a household over 47 months, with 2.1 million tuples over 7 numerical attributes. We consider ranges involving 2–7 dimensions.

- **Forest** Dua & Graff (2019): Forest cover types with 581,000 tuples and 10 numerical attributes. We consider ranges involving 2–10 dimensions.

- **IMDb** Leis et al. (2015): Information about 2.5 million movies, popular in benchmarking query optimization. While Power and Forest each have a single table, we consider multi-table join queries with range selections on columns from different tables. We also use IMDb to create drifting data distributions, as was done in Xiu et al. (2024), by "slicing" the movies by production year such that each data slice has a naturally different distribution. See details in Section D.2.

**Queries.** As there are no widely available benchmarks with drifting query distributions over real-world data, we define our own for the datasets above. For simplicity, we normalize data distributions such that every $D_t$ of interest has support in the unit hypercube $[0, 1]^d$. While our techniques generalize to arbitrary ranges in $[0, 1]^d$, we mainly restrict ourselves to orthogonal ranges because they are supported by all alternative approaches compared. To define a distribution of range queries involving $d$ dimensions, we use two parameters: a *center* $c \in [0, 1]^d$ and a *diagonal vector* $g \in [0, 1]^d$. To generate a query, we sample a center point $r$ from a normal distribution centered at $c$. Then, we sample a diagonal vector $h$ from a normal distribution centered at $g$. The resulting query is a

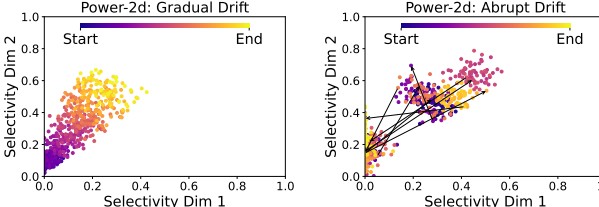

Figure D.1: Visualization of drifting query distributions. *Each axis corresponds to a selected dimension of data; each dot represents the center of a range query, with color indicating temporal progression from start to end in the query sequence $\mathcal{Z}$. To highlight significant shifts, black arrows connect consecutive queries whose centers move beyond a fixed threshold (0.3 in either dimension).*

hyper-rectangular range $[r - h/2, r + h/2]$, clipped to lie within $[0, 1]^d$. Intuitively, the center $c$ controls where queries are focused, while $g$ controls their size and aspect ratio. As a vector, $g$ provides coordinate-wise flexibility—allowing queries to be narrow in some dimensions and wide in others.

To synthesize drifting query distributions for our experiments, we construct two scenarios (further details in Section D.1):

- **Gradual drift**: In this scenario, the query distribution drifts gradually over time, from an initial distribution $Q_1$ parameterized by $(c_{\text{start}}, g_{\text{start}})$ to a final distribution $Q_n$ parameterized by $(c_{\text{end}}, g_{\text{end}})$. We ensure that the two distributions have sufficient distance in between to induce a meaningful drift. Between $Q_1$ and $Q_n$, to obtain an intermediate query distribution $Q_t$ with setting $(c_t, g_t)$, we linearly interpolate between $(c_{\text{start}}, g_{\text{start}})$ and $(c_{\text{end}}, g_{\text{end}})$. This method produces a smooth drift in both location and size/aspect ratio.[2]

- **Abrupt drift**: Here, the query distribution would remain the same for a duration of time, after which a sudden abrupt change occurs. Specifically, at fixed intervals, we sample a new center $c' \in [0, 1]^d$ and a new diagonal vector $g' \in \mathbb{R}^d_{\geq 0}$ to be used for the new query distribution, again ensuring sufficient separation from the previous setting to create a meaningful shift. For example, if the stable period is $k$, the first $k$ query distributions $Q_1, \ldots, Q_k$ will be the same, defined by a fixed $(c, g)$; then, a new setting $(c', g')$ is sampled that defines $Q_{k+1}, \ldots, Q_{2k}$, and so on. This setup induces a piecewise stationary process with sharp transitions between phrases.

Figure D.1 visualizes these two types of query drift using Power-2d workload as an example, where queries are 2-d ranges over 2 selected data dimensions. In the gradual drift case, queries evolve smoothly over time, forming a continuous trajectory. In contrast, the abrupt drift case exhibits sudden directional changes and clear spatial jumps.

**Methods compared.** We compare against other approaches that can operate using query feedback only, without requiring access to the underlying data. Learned query-driven models can be broadly categorized into two types: deep learning and distribution-based. We pick one representative of each class, along with a well-studied standard baseline:

- CDF Wu et al. (2025), a representative of the deep learning type, is a recent state-of-the-art model that has been shown to be more robust than previous work against drifts, both theoretically and empirically. The original implementation of CDF only supports one-sided ranges; we modify it to support two-sided ranges.

- PtsHist Hu et al. (2022) is a representative distribution-based model.

- MSCN Kipf et al. (2019) is a widely studied learned model that has served as a standard baseline for comparison in related work. Note that MSCN also has features that use samples from the underlining data; to ensure fair comparison in a purely query-driven setting, we turn off such features in our experiments.

While DUSS maintains a dynamic model that continuously adapts over time, the above methods, as with most existing ones in literature, are not designed for online updates; instead, they rely on periodic model retraining or fine-tuning. To ensure fair comparison, we implement various periodic

---

[2]Besides linear interpolation, we have also tried non-linear interpolation (e.g., sinusoidal or Bézier curves) as well as various configurations of the initial and final settings. The conclusions from evaluation results are consistent across these variants.

strategies for retraining/fine-tuning for these methods. These strategies not only vary the duration of the period, but also how much historical information to use in each retraining/fine-tuning step: one could use recent queries or all past queries (as long as the data distribution is stable). Finally, most models require some initial training. We give each model access to a fixed set of $s_{\text{init}}$ initial training queries along with their observed selectivities, drawn from the first workload state distribution in the sequence $\mathcal{W}$ (i.e., based on $Q_1$ and $D_1$). Algorithm performance is then evaluated on the sequence of testing queries in $\mathcal{Z}$ separate from the initial training queries. Unless otherwise specified, we set $s_{\text{init}} = 2{,}000$ for each model $\mathfrak{M} \in \{\mathsf{CDF}, \mathsf{MSCN}, \mathsf{PtsHist}\}$. We then explore the following general adaptation strategies:

- $\mathfrak{M}\text{-}S$: After initialization, the model remains static and is used to predict on $\mathcal{Z}$ without any further model updates.

- $\mathfrak{M}\text{-}R$ $(w, p)$: After initialization, the model is retrained every $p$ queries using the most recent $w$ queries. When $w = \infty$, the model is retrained on all queries seen so far.

- $\mathfrak{M}\text{-}T$ $(w, p)$: After initialization, the model is fine-tuned (as opposed to fully retrained) after every $p$ queries using the most recent $w$ queries. This strategy is supported only by $\mathsf{MSCN}$ and $\mathsf{CDF}$, where updates are performed using stochastic gradient descent on the $w$ queries for a few epochs. It is not applicable to $\mathsf{PtsHist}$, which requires solving a non-negative least-squares problem and does not support incremental updates.

**Performance metrics.** To measure predictive performance, we use several metrics. For overall accuracy, we use two standard measures: RMSE (Root Mean Squared Error) and percentile Q-Error Moerkotte et al. (2009). Given a set of $n$ test queries $\{R_1, \ldots, R_n\}$ with estimated selectivities $\hat{s}(R_i)$ and true selectivities $s(R_i)$, RMSE is defined as $(\frac{1}{n} \sum_{i=1}^{n} (\hat{s}(R_i) - s(R_i))^2)^{1/2}$; Q-error$(p)$ is defined as the $p$-th percentile of the set of relative errors:
$$\{\max\{\hat{s}(R_i), s(R_i)\} / \min\{\hat{s}(R_i), s(R_i)\} : i \in [n]\}$$
While RMSE focuses on absolute error and penalizes large deviations heavily, Q-error highlights relative error, capturing performance across varying scales of selectivity.

Finally, to be practical, a model must adapt efficiently and provide fast predictions. Therefore, we also measure the computation overhead of model retraining/fine-tuning as well as inference. All experiments were conducted on a Linux server equipped with an Intel(R) Xeon(R) Gold 5215 CPU @ 2.50GHz, 256 GB of RAM, and a NVIDIA GeForce RTX 3090 GPU (24 GB), running CUDA 12.8.

## D.1 Fixed Data, Drifting Queries

Table D.1: Selectivity estimation accuracy and training cost under gradual and abrupt query drifts on Power-2d and Power-7d workloads. ▶ *marks the lowest error or training time;* ▷ *marks the second- and third-lowest.*

| Method | Power-2d Gradual | | | | Power-2d Abrupt | | | | Power-7d Gradual | | | | Power-7d Abrupt | | | |
|---|---|---|---|---|---|---|---|---|---|---|---|---|---|---|---|---|
| | RMSE | *Med.* q-err | 90-th q-err | Train (s) | RMSE | *Med.* q-err | 90-th q-err | Train (s) | RMSE | *Med.* q-err | 90-th q-err | Train (s) | RMSE | *Med.* q-err | 90-th q-err | Train (s) |
| DUSS | ▷0.026 | ▷1.154 | ▶2.6 | ▷5 | ▶0.027 | ▷1.055 | ▶1.8 | ▶5 | ▶0.092 | ▶1.364 | ▷14.9 | ▷9 | ▶0.072 | ▶1.215 | ▶17.9 | 22 |
| CDF-R (∞, 2k) | 0.105 | 2. | 24.9 | 236 | 0.242 | 11.096 | 4410.0 | 245 | 0.195 | 2.397 | 22.4 | 296 | 0.215 | 3.196 | 126.1 | 288 |
| CDF-R (∞, 500) | 0.063 | 1.439 | 4.0 | 1176 | 0.176 | 1.556 | 237.0 | 1224 | 0.139 | 1.679 | ▷11.0 | 1361 | 0.164 | 3.221 | ▷60.0 | 1309 |
| CDF-R (2k, 2k) | 0.095 | 1.761 | 12.0 | 255 | 0.363 | 8. | 4410.0 | 277 | 0.179 | 2.471 | 33.0 | 290 | 0.221 | 3.330 | 2163.0 | 288 |
| MSCN-R (∞, 2k) | 0.099 | 1.579 | 12.0 | 49 | 0.201 | 3.042 | 4410.0 | 50 | 0.171 | 2. | 23.9 | 52 | 0.215 | 4.442 | 217.3 | 55 |
| MSCN-R (∞, 500) | 0.071 | 1.435 | 4.8 | 232 | 0.145 | 1.613 | 45.0 | 252 | 0.148 | 1.525 | ▷7.0 | 248 | 0.168 | 2.446 | ▷47.6 | 253 |
| MSCN-R (2k, 2k) | 0.102 | 1.557 | 12.0 | 19 | 0.273 | 458. | 4410.0 | 14 | 0.217 | 3. | 50.5 | 20 | 0.263 | 10.221 | 1998.0 | 25 |
| PtsHist-R (∞, 2k) | ▷0.036 | 1.178 | 4.6 | 135 | ▷0.135 | ▷1.027 | 1709.0 | 150 | 0.106 | 1.484 | 38.2 | 131 | ▷0.104 | ▷1.340 | 153.7 | 134 |
| PtsHist-R (∞, 500) | ▶0.016 | ▶1.110 | ▷3.4 | 632 | ▷0.086 | ▶1.017 | ▷6.6 | 711 | ▷0.101 | ▷1.469 | 35.0 | 607 | ▷0.095 | ▷1.296 | 75.9 | 616 |
| PtsHist-R (2k, 2k) | ▷0.036 | ▷1.116 | ▷3.7 | 73 | 0.289 | 626. | 4612.0 | 79 | ▷0.103 | ▷1.456 | 36.6 | 76 | 0.210 | 3.745 | 2584.0 | 80 |
| CDF-T (2k, 2k) | 0.099 | 2. | 24.1 | 54 | 0.271 | 269. | 4607.0 | 51 | 0.197 | 2.541 | 41.0 | 68 | 0.257 | 9. | 241.5 | 72 |
| CDF-T (500, 500) | 0.058 | 1.501 | 6.0 | 109 | 0.181 | 2. | 2765.0 | 108 | 0.163 | 2.124 | 21.5 | 123 | 0.197 | 3.157 | 283.5 | 121 |
| MSCN-T (2k, 2k) | 0.147 | 2.141 | 23.0 | 10 | 0.279 | 345.471 | 4227.0 | 6 | 0.189 | 2.112 | 38.1 | 10 | 0.201 | 3.502 | 105.1 | ▶5 |
| MSCN-T (500, 500) | 0.071 | 1.599 | 8.2 | 12 | 0.198 | 2.269 | 618.1 | 8 | 0.133 | 1.846 | 10.9 | 12 | 0.195 | 3.002 | 120.4 | ▷10 |
| DUSS-S | 0.149 | 5.848 | 22.4 | ▶3 | 0.224 | 6.149 | ▷40.3 | ▶3 | 0.203 | 2.808 | 130.2 | ▷6 | 0.110 | 1.366 | 159.1 | 12 |
| CDF-S | 0.181 | 10. | 4000.0 | 15 | 0.260 | 53.5 | 4643.0 | 24 | 0.320 | 20.330 | 146.0 | 24 | 0.250 | 9.210 | 67.2 | 18 |
| MSCN-S | 0.181 | 11. | 4076.0 | ▶3 | 0.275 | 20.280 | 4608.0 | ▶3 | 0.310 | 15.554 | 116.9 | ▶2 | 0.215 | 4.749 | 64.7 | ▶5 |
| PtsHist-S | 0.137 | 797. | 797.0 | 13 | 0.239 | 9.121 | 4609.0 | 14 | 0.177 | 2.491 | 138.0 | 13 | 0.113 | 1.580 | 200.0 | 13 |

We first consider the simpler setting where the data distribution remains fixed, while the query distribution undergoes drift. Although this setting is "simpler," it still presents considerable challenge for

learners relying only on query-feedback. We focus on single-table range selection queries over the Power and Forest. For each dataset, we compare competing query-driven approaches under multiple workloads, with the number of dimensions in the query range varying from 2 to 7, and query drift being either gradual or robust. The change in range dimensionality is intended to evaluate the models under varying degrees of query complexity. Each workload consists a sequence of $n = 10{,}000$ testing queries, separate from the initial $s_{\text{init}}$ training queries.

**Results on Power.** Table D.1 presents results for Power-2d and Power-7d under both gradual and abrupt drift scenarios. To help spot top performers, we highlight the best values in each column representing a performance metric. Note that the reported training times ("Train" column) broadly include model initialization using the $s_{\text{init}}$ training queries, as well as all subsequent updating, re-training, or fine-tuning costs incurred while processing the testing workload.

As we can see from Table D.1, DUSS consistently delivers the best trade-off between accuracy and efficiency. It ranks among the top three for all accuracy metrics across workloads, while incurring some of the lowest training times. In contrast, competing methods occasionally match or slightly surpass its accuracy, but only do so by incurring substantially higher training costs. For example, for Power-2d under gradual drift, PtsHist achieves marginally lower RMSE than DUSS, but requires over $120\times$ more training time. As the query drift intensifies—particularly in high-dimensional or abrupt settings—the advantage of DUSS becomes even more pronounced: it is the only method to maintain superior accuracy with minimal training costs—typically under 22 seconds in total. (To put this number in context, it represents a mere 2% of the time needed to execute all queries in the workload.)

These results also shed light on the effectiveness of different adaptation strategies. Static baselines, while requiring no further cost to maintain, consistently underperform in accuracy, highlighting the importance of model adaptability in a dynamic setting, even if only the query distribution drifts (and the data distribution does not). Among the adaptive variants, $\mathfrak{M}\text{-}R\,(\infty, 500)$—which retrains on the full query history at high frequency—typically delivers the highest accuracy, but incurs substantial training time. Reducing the frequency to $\mathfrak{M}\text{-}R\,(\infty, 2000)$ lowers cost, though at the expense of accuracy. Fine-tuning strategies like $\mathfrak{M}\text{-}T\,(500, 500)$, which incrementally update the model using fewer epochs, strike a middle ground: they lower overhead compared to full retraining while improving accuracy over infrequent retraining. However, they still cannot match the best-performing retrained models, and they remain more costly than DUSS.

Moreover, we note that the right adaptation strategy depends heavily on the drift scenario. Under gradual drift, retraining on only recent queries (e.g., $\mathfrak{M}\text{-}R\,(2\text{k}, 2\text{k})$) is often sufficient and cost-efficient. However, this approach performs poorly under abrupt drift—sometimes it is even worse than a static model—as it neglects earlier but still relevant queries. In such cases, retraining on the full query history, as in $\mathfrak{M}\text{-}R\,(\infty, 2\text{k})$, proves more robust and reliable. In contrast, DUSS does not have this problem of having to pick the right adaptation strategy at all.

**Additional results on Power and Forest.** Complementing Table D.1, Figure D.2 visualizes the trade-off between accuracy (RMSE) and maintenance cost (log-scaled training time) for Power-2d/7d and Forest-2d/10d under both gradual and abrupt drift scenarios. The trade-offs achieved by different variants of the same approach are connected into one curve. For DUSS, we additionally consider a "static" variant where we freeze its model after initiation and prevent it from dynamic adaption; the performance of this variant is then connected to the normal DUSS. Other approaches are shown under three retraining strategies with varying retraining frequencies: $\mathfrak{M}\text{-}S$, $\mathfrak{M}\text{-}R\,(\infty, 2\text{k})$, and $\mathfrak{M}\text{-}R\,(\infty, 500)$.

DUSS's ability to achieve high accuracy with minimal maintenance cost is clearly illustrated in the figure. Across all scenarios, including high-dimensional and abrupt drift cases, DUSS achieves strong accuracy with under 30 seconds of training time. In comparison, competing methods require significantly more time to reach similar performance. PtsHist-$R\,(\infty, 500)$ is the most competitive among the baselines in terms of accuracy, especially under gradual drift in low-dimensional settings. However, it still falls short of DUSS under abrupt drift and incurs significantly higher cost, up to 1,000 times more in cases like Forest-2d. CDF is often more accurate than MSCN (but not always); at the same time, it is more costly.

Table D.2: Average total training time and end-to-end inference time (per query) comparison across methods.

|  | DUSS | CDF-$R$ ($\infty$, 500) | MSCN-$R$ ($\infty$, 500) | PtsHist-$R$ ($\infty$, 500) |
|---|---|---|---|---|
| Train (s) | 12 | 1363 | 236 | 1603 |
| Inference (ms) | 0.4 | 1.5 | 1.4 | 11.9 |

(a) Power-2d (gradual)     (b) Power-2d (abrupt)     (c) Power-7d (gradual)     (d) Power-7d (abrupt)

(e) Forest-2d (gradual)     (f) Forest-2d (abrupt)     (g) Forest-10d (gradual)     (h) Forest-10d (abrupt)

Figure D.2: RMSE vs. log-scaled training time (in seconds) for different estimators under gradual and abrupt query drift.

**Inference cost.** Last but not least, we measure the average end-to-end inference time per query for different models across various scenario enumerated in Figure D.2. Results are shown in Table D.2. Inference speed is a critical factor in assessing the practicality of a selectivity estimator, as slower inference slows down query optimization and prolongs end-to-end query latency. Thanks to its simple model, DUSS, even with a straightforward implementation, achieves the lowest inference time among all approaches. Deep-learning approaches require preprocessing steps such as zero-padding, mask generation, and tensor conversion to ready each incoming query for estimation. Both DUSS and deep-learning approaches offer reasonable inference speed, typically under 1.5ms per query. In contrast, PtsHist incurs significantly higher inference overhead due to its more complex internal structure and geometric computations. Although PtsHist can occasionally outperform DUSS in accuracy after nontrivial training efforts, its high inference cost limits its suitability for latency-sensitive scenarios.

### D.2 DRIFTING QUERY AND DATA DISTRIBUTIONS

We now consider a more challenging and realistic scenario where both the query and data distributions are drifting simultaneously over time. Table 5.1 presents the results for four workloads based on IMDb. The queries in IMDb-2d come from a 2-way join query template, with local range selections on both tables; those in IMDb-7d come from a 7-way join query template, with local range selections on 6 tables. Changes in the distribution of query ranges are generated as described earlier in this section. To simulate changes in data distribution, we partition the IMDb dataset based on the production year of the movies (*title.production_year*). Specifically, we define five slices: 2015-2006, 2012-2003, 2009-2000, 1999-1981, and 1980-1880, denoted $D_1$ to $D_5$, respectively. Each slice includes the movies produced within the corresponding year range, along with associated data in other tables, and is treated as a standalone database instance. The first three instances cover recent movies in a sliding-window fashion: each slice is 10 years and overlaps with adjacent slices by 7 years. The last two slices include older movies, with no overlap with the earlier slices. In effect, the sequence simulates somewhat gradual data distribution shifts between $D_1$ to $D_3$, followed by more abrupt and significant changes to $D_4$ and $D_5$. For the query workload, we again consider the two types of query drifts studied in Section D.1, gradual and abrupt. Each query workload contains $n = 50{,}000$ queries, divided into equal-sized chunks and assigned to the five corresponding data slices in order.

For each competing query-driven method $\mathfrak{M}$, we evaluate two retraining strategies: $\mathfrak{M}$-$R$ (2k, 2k) and $\mathfrak{M}$-$R$($\infty$, 2k). In the $\mathfrak{M}$-$R$($\infty$, 2k) setup, instead of using all historical queries to retrain, we restrict them to queries that were executed against the current database slice (if they are available). This restriction

is intuitive because earlier queries would have provided incorrect selectivity feedback. On the other hand, the knowledge about when the underlying database slice has changed is in fact unavailable to the model; therefore, we are effectively giving this setup an unfair advantage over others.

From Table 5.1, we observe trends consistent with those in Section D.1, despite the added difficulty of simultaneous shifts in both data and query distributions. First, DUSS consistently achieves the best accuracy and training efficiency across both gradual and abrupt query drift settings, outperforming all other query-driven methods—including PtsHist, the strongest among them. Second, as before, retraining on all observed queries yields better accuracy than using only recent windows—particularly under abrupt query drifts, where relying solely on recent queries can be detrimental. Third, CDF generally achieves lower RMSE than MSCN, but this improvement comes at the cost of significantly higher training time.

In Figure 5.1, we track RMSE and Q-error using a sliding window of size 100 on IMDb-7d with abrupt query drift—our most challenging setting. Each metric reflects the average error within a window. A well-adapting model should sustain low error even with small windows. We compare DUSS against PtsHist-R $(\infty, 2k)$, the most competitive baseline in Table 5.1. As expected, both models exhibit error spikes around distribution shifts—such as transitions between data slices (e.g., from $D_n$ to $D_{n+1}$) and query shifts at $15K$ and $45K$. However, DUSS adapts more quickly and returns to good accuracy sooner. For RMSE, it has lower worst-case error and faster recovery, whereas PtsHist lags behind even with much heavier retraining. For Q-error, DUSS remains consistently lower with smaller fluctuation. These results further demonstrate DUSS's ability to adapt to both data and query distribution shifts.

# E  ADDITIONAL RELATED WORK

**Cardinality Estimation.** Cardinality/Selectivity estimation is a fundamental problem in query processing Lipton et al. (1990); Poosala & Ioannidis (1997); Aboulnaga & Chaudhuri (1999); Bruno et al. (2001); Srivastava et al. (2006); Markl et al. (2007); Kaushik & Suciu (2009). Recently, there has been significant interest in ML-based techniques for selectivity estimation Park et al. (2020); Hasan et al. (2020); Kipf et al. (2019); Yang et al. (2020); Dutt et al. (2019); Hilprecht et al. (2020); Wang et al. (2021). Broadly, ML-based approaches falls into two categories, including *data-driven* Hilprecht et al. (2020); Yang et al. (2020); Hasan et al. (2020) and *query-driven* Park et al. (2020); Kipf et al. (2019); Hu et al. (2022); Wu et al. (2025); Dutt et al. (2019). Data-driven methods aim to model the underlying data distribution by directly accessing full tables or sampled subsets. In contrast, query-driven methods focus on specific query workloads and typically learn from query–selectivity feedback. A large variety of models has been proposed for the query-driven setting, including methods based on probability distributions (e.g., histograms, mixture models), tree ensembles, graphs and deep neural networks. For a comprehensive survey, see Wang et al. (2021).

Several strategies have been adopted to handle query and data drift. For example, Robust-MSCN Negi et al. (2023) extends basic MSCN Kipf et al. (2019), leveraging up-to-date DBMS statistics and data sampling-based features. On the other hand, CDF-MSCN Wu et al. (2025) shows that distribution-based models are robust against query drift and modifies MSCN to have this property. However, when the drift is enough, no fixed model can perform well without retraining/finetuning. This has led to sophisticated techniques such as Warper Li et al. (2022), which employs a Generative Adversarial Network (GAN) to synthesize additional training queries. More recently, ShiftHandler Wu & Ives (2024) proposes a replay buffer to select a smaller, high-impact subset of training queries for retraining. DDUp Kurmanji & Triantafillou (2023) considers how to update models in the presence of data updates. Typically, methods used to maintain an accurate model in fully dynamic environments require data access. This is in contrast to our algorithm DUSS which only works based on observtaions of user generated queries and their cardinalities. For example, Warper Li et al. (2022) and ShiftHandler Wu & Ives (2024) accesses data by re-executing queries; Robust-MSCN Negi et al. (2023) rebuilds its sample bitmaps.

**Learning Under Drift.** The general problem of learning under drift has been extensively studied in the machine learning community and giving a full overview is beyond our scope. Early work by Helmbold & Long (1994) established learning bounds under the assumption that only the target concept may drift. Subsequently, significant extensions were made by Bartlett (1992); Barve & Long

(1996). We use the framework by Mohri & Medina (2012), who themselves extended the results of Bartlett (1992) to real-valued functions.

## F   LLM DECLARATION

All intellectual contributions in this paper are solely due to the authors. We made limited use of a large language model (ChatGPT) to edit prose. Specifically, certain paragraphs written by the authors were lightly polished for style, grammar and clarity. The model was not involved in the generation of research ideas, including algorithms, proofs, or design of experiments. AI-assistance was used for coding our algorithms, especially for debugging.

