# OpenReview forum: "Learning Range-Query Selectivity under Drifting Query and Data Distributions with Provable Bounds"
_ICLR.cc/2026/Conference — ICLR 2026 Conference Withdrawn Submission_

### Official Review · Reviewer_VwBt · 2025-10-28

**Soundness:** 2
**Presentation:** 1
**Contribution:** 2
**Rating:** 0
**Confidence:** 2

**Summary:**

=====

I would like to clarify that though I have some ML theory background, I was not familiar with the setting and goal described in this work.

Despite carefully reading this work as well as some of the referenced sources, I still don't feel like I have a clear understanding of its setting.

As such, my feedback will focus on pointing out the confusing points and proposing possible presentation improvements, when sharing this work with ML audience.

=====

This work is concerned with selectivity-estimation, which seems to combine elements from statistical learning and distribution estimation. In this setting, we consider a domain of elements $X$, and a family subsets of $\mathcal{X}$ denoted by $\mathcal{R}$ (these are effectively binary functions). A distribution $D$ over $\mathcal{X}$ induces a \emph{selectivity function} $\mu_{D}: \mathcal{R} \to [0,1]$ which assigns to each subset $R \in \mathcal{R}$ its probability mass under $D$ (its expectation, in terms of functions). A function $h: \mathcal{R} \to [0,1]$ is called a hypothesis. Given an additional distribution $Q$ over $\mathcal{R}$, we define the error of $h$ w.r.t. $Q$ as the expected $\vert h(R) - Q(R)$ where $R \sim Q$. Following the classical learning definition, a learning algorithm receives a sequence of samples sampled from $Q$ (in this case, ranges and their selectivities) and produces a hypothesis, and its goal is to minimize the expected error of the hypothesis w.r.t. $Q$.

The authors provide improved bounds when $\mathcal{R}$ "correspond to geometric objects of constant size" (I am still not sure what this means), then continue to generalize these results to settings where the data distribution $D$ and / or ranges distribution $Q$ dynamically change.

**Strengths:**

Unfortunately, I did not manage to gain sufficient understanding to be able to attest for this work's strengths.

**Weaknesses:**

I found the presentation very confusing. I will provide some examples in this section.

This work moves back and forth between many distributions, and I have failed to understand their relation. Distribution over elements is denoted by $D$, but in other parts the authors consider a dataset $C$ which induces a uniform distribution over a subset of elements. Crucially, this dataset is not the training data for the algorithm, and its purpose is hard to explain.
Additionally, the authors consider a family of data distributions $\mathcal{D}$, but I have failed to understand how it relates to the definition of learnability or the stated theorems.

Many statements and definition are not formally defined, which makes it hard to follow. For example, from the verbal description it seems like learnability is defined w.r.t. any distribution $W$  (product of $D$ and $Q$). On the other hand, the family of hypotheses (denoted by $\mathcal{M}_{\Sigma, \mathcal{D}}$) is defined w.r.t. $\mathcal{D}$ - a family of distributions over $\mathcal{X}$.
 Even more problematic is the statement of Theorem 2.1 which is defined with respect to a family of ranges "correspond to geometric
objects of constant size such as boxes, or balls or halfspaces." The term "constant size" is not defined in this context, and from looking at the appendix it seems to be related to some notion of VC dimension of $\mathcal{R}$ with respect to $\mathcal{X}$ (at least that's what I think) which was not defined in the body of the paper.

There are many other smaller confusing issues such that the usage of nearly identical fonts for the set of ranges and a single range in it, the usage or the term "finite multiset of tuples" to describe elements in the domain (what are the tuples?), inconsistent usage of calligraphic, bold, and other visual aids, alongside introduction of a large number of symbols without a clear structure,

Finally, I have failed to verify even the first theorem. As mentioned, its statement is ill defined, the body does not contain even a proof outline, and the appendix includes only a short hand-wavy of the result.

**Questions:**

I don't have the capacity to form concrete questions at this point.

---

### Official Review · Reviewer_aB1o · 2025-10-31

**Soundness:** 3
**Presentation:** 3
**Contribution:** 3
**Rating:** 4
**Confidence:** 3

**Summary:**

Proposes and analyzes a method to perform selectivity estimation with absolute error bounds on (certain) range queries when (i) only query-selectivity results but no access to the data is available, (ii) the query distribution may drift, (iii) the data distribution may drift.

From an ML standpoint, the proposed method is conceptually simple and, as the various theoretical results prove, has low regret under (ii) as well as low error under (iii) when the drift is slow. The results use various techniques from learning theory and are sound, interesting, well-presented. From a database systems perspective, the paper falls short: the analysis omits query-dependent terms (W1), the method is not suitable for its intended use case (W2), and the experimental study is too limited (W3).

**Strengths:**

S1. Simple method

S2. Analysis provides error guarantees

S3. Purely observational (no data access required)

S4. Query and data distribution drift handled under certain assumptions

**Weaknesses:**

W1. Analysis limited. A key drawback of the analysis is that the method assumes a fixed set of possible range queries (R in paper), but does not account for its properties in the complexity analysis. In fact, at its heart, the method uses a large random sample from the data space (set of all possible tuples) and increments/decrements sample weights when estimates overshoot/undershoot. The required size of this sample drives computational costs, but it is not analyzed but hidden in the O-notation. I'd expect it to excessively large (e.g., exponential in the number of dimensions) unless the boxes/spheres have substantial area in each dimension and data is low-dimensional.

From now on, I take a database systems standpoint, as the authors describe query selectivity estimation as the intended application of their method.

W2. Not suitable for query selectivity estimation. This is for a number of reasons, the first one being the one described above under W1.

(i) From a cost standpoint, query selectivity estimation needs to be fast, and running a query on a large sample is too costly (and may, in fact, require query optimization as well).

(ii) From a query standpoint, the set of range queries supported by small sample sizes is likely to be too limiting.

(iii) From an error perspective, q-error is much more suitable than absolute error.

(iv) Finally, the novelty of the method is in disallowing access to data. As soon as data access is allowed (as in virtually all DBMS'), the method becomes uninteresting in practice.

W3. Experimental study too limited.

(i) It's very likely that the authors are aware of all of the points I made above: in the experimental study, the authors modify their method to dynamically create a dynamic sample (but don't specify how exactly it is maintained). While I can see that one can build such a method without pruning and while maintaining the error guarantees, details are missing here. As soon a pruning/compression is used, the error guarantees likely fall apart.

(ii) The authors use a too-simple synthetic query distribution: all queries are essentially the same local range + noise + slow drift. That's great for their approach in (i), but does not allow for a fair assessment. In particular, the statement that the method uses little space is highly misleading, as are its accuracy results. Queries should be designed such that all of tuples of the actual data are covered by both small as well as large selectivity queries.

(iii) For the fixed data distribution setting, a comparison to common selectivity estimators is missing. Sure, they use data, but showing results would allow to judge how good the estimates produced here really are. The perhaps simplest baseline would be a random sample of the actual data and of the same size as the support set (and random samples can be produced on-the-fly or maintained by DBMS when the data changes).


Minor:

If a random sample is used to obtain the support set, then all results only hold with high probability.

232: Chi not introduced (only appendix).

244: Count not introduced (only appendix).

**Questions:**

-

---

### Official Review · Reviewer_YR1C · 2025-11-01

**Soundness:** 3
**Presentation:** 2
**Contribution:** 3
**Rating:** 4
**Confidence:** 4

**Summary:**

The paper considers the problem of cardinality estimation of queries under both data and query shift. Under the setup considered in this work, a learner gets to see only the query-cardinality pair at each step, after outputting the estimate at each time. This work considers modeling this problem as an online learning problem over changing data distribution and query distribution as sequence of state distributions where each query and data is sampled from distribution at that time respectively. The authors consider two types of error: tracking where error from output at each time step is small and regret where the number of times algorithm makes large error is small. One of the main results in this work is improved sample complexity bound for learning selectivity functions. This improved bound is used in the analysis of their algorithm to obtain $\varepsilon$ tracking error when the change is distributions at each step is very small $o(\varepsilon^3)$, and $O(1/\varepsilon^3)$ regret when data distribution is either fixed or stable but query distribution can vary arbitrarily.

**Strengths:**

The problem is well motivated and results for both types of error normally considered in online learning are provided.
The improved sample complexity bound forms the core of their results. This is them used to design a sliding window type algorithm, where a distribution over the data space is maintained and update using multiplicative weight type updates. The experimental results show that algorithm stays relatively low cost even with abrupt changes.

**Weaknesses:**

The algorithm requires maintaining a set $S$ that is representative enough across all data distributions. For range queries maintaining a grid is a natural way, but doing this generally and for when the support of data might change with the distribution for other kinds of queries seems challenging. For tracking error to be small, the necessary change in distribution is very small $O(\varepsilon^{-3})$.

**Questions:**

The problem is described as learner only being allowed to see the query-answer pair, but in reality is also provided with an $\varepsilon$-representative set of the data. There seems to be mismatch in the description, could the authors please elaborate on the assumption.

---

### Official Review · Reviewer_uVh1 · 2025-11-03

**Soundness:** 2
**Presentation:** 3
**Contribution:** 3
**Rating:** 4
**Confidence:** 3

**Summary:**

The paper studies the problem of learning cardinality estimation function under data and query drift. It specifically formulates the problem as an online learning problem and proposes a solution to solve it. The paper provides theoretical bounds on the quality of the solution in terms of regret and shows the method is a tracking algorithm. Experimental results show the method achieves low error.

**Strengths:**

- The formulation of the cardinality estimation problem as an online learning problem is interesting

- The paper shows comprehensive results on quality of the method on various drift scenarios.

- There is empirical validation of the bounds

**Weaknesses:**

- The paper should discuss provide a more detailed comparison with Wu et al. (2025) that also considers both query and distribution shift. The paper says "However, their results do not extend to the fully dynamic setting, where both query and data distributions evolve.", but the should explicit state what settings their paper considers that Wu et al. (2025) doesn't.

- Although I think the online learning formulation is interesting, I don't understand the significance of the results. Consider the setting with finite X. Then |S| will be more than the size of the database, and using it to estimate cardinality is pointless. Overall, with the construction provided, it seems like it's less computationally expensive to use the database to obtain exact answers rather than actually use the model to make an estimate

- Relatedly, there is no discussion on time/space complexity of using the method to obtain an estimate.  Overall, for cardinality estimation, low error on it's own is not important, but rather it should be put in the context of time/space complexity

- For theorem 2.1 why is there no dependence on what model M is used? I'd expect it to makes a difference what model is used (e.g., whether it's a neural network or histogram)

**Questions:**

Please provide a discussion on time and space complexity of the method, and how it compares with with obtaining exact answers by running the query on the database; also respond to the other points raised above.

---

### Note · Authors · 2025-12-04

I have read and agree with the venue's withdrawal policy on behalf of myself and my co-authors.